# Endothelial MT1-MMP targeting limits intussusceptive angiogenesis and colitis via TSP1/nitric oxide axis

Sergio Esteban[1], Cristina Clemente[1,2], Agnieszka Koziol[1], Pilar Gonzalo[1], Cristina Rius[1,3], Fernando Martínez[4], Pablo M Linares[5], María Chaparro[5], Ana Urzainqui[6], Vicente Andrés[1,3] (iD), Motoharu Seiki[7], Javier P Gisbert[5] & Alicia G Arroyo[1,2,*] (iD)

## Abstract

Pathological angiogenesis contributes to cancer progression and chronic inflammatory diseases. In inflammatory bowel disease, the microvasculature expands by intussusceptive angiogenesis (IA), a poorly characterized mechanism involving increased blood flow and splitting of pre-existing capillaries. In this report, mice lacking the protease MT1-MMP in endothelial cells (MT1$^{i\Delta EC}$) presented limited IA in the capillary plexus of the colon mucosa assessed by 3D imaging during 1% DSS-induced colitis. This resulted in better tissue perfusion, preserved intestinal morphology, and milder disease activity index. Combined *in vivo* intravital microscopy and lentiviral rescue experiments with *in vitro* cell culture demonstrated that MT1-MMP activity in endothelial cells is required for vasodilation and IA, as well as for nitric oxide production via binding of the C-terminal fragment of MT1-MMP substrate thrombospondin-1 (TSP1) to CD47/αvβ3 integrin. Moreover, TSP1 levels were significantly higher in serum from IBD patients and *in vivo* administration of an anti-MT1-MMP inhibitory antibody or a nonamer peptide spanning the αvβ3 integrin binding site in TSP1 reduced IA during mouse colitis. Our results identify MT1-MMP as a new actor in inflammatory IA and a promising therapeutic target for inflammatory bowel disease.

**Keywords** inflammatory bowel disease; intussusceptive angiogenesis; MT1-MMP; nitric oxide; TSP1
**Subject Categories** Digestive System; Immunology; Vascular Biology & Angiogenesis

See also: **G D'Amico** *et al* (February 2020)

## Introduction

The vasculature delivers oxygen and nutrients to all tissues and must constantly and dynamically adapt to tissue needs. Angiogenesis, the formation of new capillaries from pre-existing vessels, is required to expand the vasculature not only during development and tissue repair, but also in pathological conditions such as cancer and chronic inflammatory disease (Potente *et al*, 2011). In these contexts, the newly formed vasculature is often marginally functional and leaky, contributing to disease progression (De Bock *et al*, 2011; Parma *et al*, 2017). Angiogenesis often occurs by capillary sprouting, which is mainly triggered by hypoxia and the subsequent production of vascular endothelial growth factor (VEGF). New therapeutic interventions in cancer and other diseases have therefore focused on inhibiting sprouting angiogenesis, mostly by blocking VEGF (Potente *et al*, 2011). However, the development of resistance to this approach and its overall limited success have shifted attention to the possible existence of alternative modes of capillary expansion (Ribatti & Djonov, 2012).

Intussusceptive angiogenesis (IA) was formally recognized in the 1980s and involves the expansion of the microvasculature through the formation of intraluminal pillars, eventually resulting in capillary splitting (Burri *et al*, 2004). IA contributes to the physiological expansion of capillary beds during embryonic development and postnatal coronary vasculature remodeling (van Groningen *et al*, 1991; Djonov *et al*, 2000). IA also occurs in certain cancers and chronic diseases such as bronchopulmonary dysplasia, in which IA generates an aberrant and dysfunctional vasculature that may contribute to disease progression (Ribatti & Djonov, 2012; Giacomini *et al*, 2015; De Paepe *et al*, 2017). Sprouting and intussusception angiogenesis mechanisms can co-exist in pathophysiological settings (Konerding *et al*, 2012; Karthik *et al*, 2018).

1 Vascular Pathophysiology Area, Centro Nacional de Investigaciones Cardiovasculares (CNIC), Madrid, Spain
2 Centro de Investigaciones Biológicas (CIB-CSIC), Madrid, Spain
3 CIBER de Enfermedades Cardiovasculares (CIBER-CV), Madrid, Spain
4 Bioinformatics Unit, Centro Nacional de Investigaciones Cardiovasculares (CNIC), Madrid, Spain
5 Gastroenterology Unit, Hospital Universitario de La Princesa, Instituto de Investigación Sanitaria Princesa (IIS-IP), Centro de Investigación Biomédica en Red de Enfermedades Hepáticas y Digestivas (CIBER-EHD), Universidad Autónoma de Madrid, Madrid, Spain
6 Immunology Department, FIB-Hospital Universitario de La Princesa, Instituto de Investigación Sanitaria Princesa (IIS-IP), Madrid, Spain
7 Division of Cancer Cell Research, Institute of Medical Science, University of Tokyo, Tokyo, Japan
*Corresponding author. Tel: +34 91 837 31 12; Fax: +34 91 536 04 32; E-mail: agarroyo@cib.csic.es

IA is a dynamic, fast, and metabolically undemanding process that barely involves proliferation but instead progresses through intraluminal endothelial cell rearrangements (Burri *et al*, 2004). IA is usually driven by a persistent increase in blood flow, and it aimed to restore shear stress in the split vessels (Styp-Rekowska *et al*, 2011). However, knowledge is scarce about the cellular and molecular mechanisms underlying IA. This is partly due to the absence of *in vitro* models of IA and the limited experimental techniques to identify and quantify genuine IA events *in vivo*, apart from scanning electron microscopy and corrosion casts (Nowak-Sliwinska *et al*, 2018). Nevertheless, genes whose expression is enriched during IA have been identified in skeletal muscle, in which the vasodilator prazosin induces IA and the excision of the agonist muscle, sprouting angiogenesis (Zhou *et al*, 1998; Egginton, 2011). Upregulated genes in skeletal muscle of prazosin-treated mice included endothelial nitric oxide synthase (eNOS) and neuropilin-1 (Nrp1), suggesting a role for these pathways in IA. Analysis of eNOS (*Nos3*)-deficient mice confirmed that eNOS is required for IA but not for sprouting angiogenesis in skeletal muscle (Williams *et al*, 2006); more recently, nitric oxide (NO) has been shown to contribute to pathological IA in tumors (Vimalraj *et al*, 2018). Advanced microscopy techniques and increasing knowledge about endothelial cell responses to blood flow have together favored the recent characterization of the IA modulators endoglin and ephrinB2/EphB4 (Hlushchuk *et al*, 2017; Groppa *et al*, 2018). However, it remains unclear how these pathways are regulated and contribute to IA, particularly during disease.

IA is the mechanism of capillary expansion during intestinal inflammation, and analysis of chemically induced murine colitis (e.g., with dextran sodium sulfate; DSS) has advanced knowledge about the morphogenesis and hemodynamics underlying inflammatory IA (Mori *et al*, 2005; Ravnic *et al*, 2007; Filipovic *et al*, 2009; Konerding *et al*, 2010). These studies show that mechanical forces and changes in intraluminal blood flow drive IA and that a marked vasodilation occurs during the first stages of IA during colitis, before complete duplication of the mucosal plexus (Filipovic *et al*, 2009; Konerding *et al*, 2010). The DSS mouse model of colitis recapitulates some of the features of human inflammatory bowel disease (IBD), a chronic inflammatory disease of the intestine comprising ulcerative colitis and Crohn's disease and characterized by phases of remission and relapse (Podolsky, 2002). IBD is a multifactorial disease featuring a primary defect in intestinal epithelial barrier integrity and an exacerbated immune response to the microbiota and for which there is as yet no universal and efficient therapy (Gyires *et al*, 2014). As in other chronic inflammatory diseases, colitis progression is believed to involve angiogenesis, which is therefore regarded as a potential treatment target for human IBD (Chidlow *et al*, 2006; Koutroubakis *et al*, 2006; Danese, 2007). Attempts have been made to reduce angiogenesis and reduce colitis symptoms by targeting diverse molecular pathways, such as VEGF, TSP1/CD36, and CD40-CD40L, with limited success (Danese *et al*, 2007a,b; Punekar *et al*, 2008; Scaldaferri *et al*, 2009). There is therefore a need to decipher the molecular pathways involved in colitis-associated IA in order to design more rational therapeutic strategies.

During colitis, epithelial and mesenchymal cells, as well as other cell types, increase expression of MT1-MMP (Pender *et al*, 2000; te Velde *et al*, 2007; Alvarado *et al*, 2008), a membrane-anchored matrix metalloproteinase whose activity contributes to sprouting

angiogenesis *in vitro* and *in vivo* through the combined processing of substrates such as TSP1, NID1, and CYR61 (Galvez *et al*, 2001, 2005; Koziol *et al*, 2012a). MT1-MMP also has non-proteolytic actions that contribute to the regulation of Rac1 or HIF1α signaling (Koziol *et al*, 2012b). MT1-MMP is expressed in endothelial cells (ECs) at low levels in homeostatic conditions, but it is upregulated by the pro-inflammatory cytokines TNFα and interleukin-1 (Rajavashisth *et al*, 1999; Koziol *et al*, 2012a). Despite the important role of MT1-MMP in sprouting angiogenesis, the potential contribution of endothelial MT1-MMP to IA, particularly in the context of inflammation and IBD, has not been explored previously.

In the present study, the analysis of mice specifically lacking MT1-MMP in ECs identified this protease as an actor in inflammation-driven IA whose endothelial targeting results in preserved vasculature and amelioration of colitis. Deciphering the underlying MT1-MMP/TSP1/αvβ3 integrin/NO molecular axis opens avenues for the development of new diagnostic and therapeutic interventions in IBD.

# Results

## MT1-MMP is required for intussusceptive angiogenesis (IA) in DSS-induced colitis

To investigate the possible role of MT1-MMP (gene name *MMP14*) in inflammation-driven IA, we used the DSS-induced colitis model, a widely recognized model of IA (Konerding *et al*, 2010). We first assessed MT1-MMP expression in ECs of the mucosa vasculature in the colon by tracking β-gal expression in MT1$^{lacZ/+}$ reporter mice. In non-treated mice, nuclear β-gal expression was present in ECs in the vessels nearby the muscularis mucosa but was barely detected in those of the mucosal plexus, the polygonal capillary network around the colonic crypts. Treatment of mice with 1% DSS significantly increased the proportion of endothelial cells expressing β-gal in the mucosal plexus after 3 days with a remaining augmented trend at 7 days (Appendix Fig S1A and B). Patches of β-gal-positive ECs were frequently detected near the Y-junctions (tri-corners) in the mucosal vascular plexus (Appendix Fig S1C). MT1-MMP was deleted in the ECs of MT1-MMP$^{f/f}$;*Cdh5*Cre$^{ERT2}$ mice (MT1$^{iΔEC}$) mice by daily injections of 4-hydroxy-tamoxifen (4-OHT) for 5 days; 4-OHT injections began 3 days before DSS treatment. Isolated lung ECs were examined to confirm efficient recombination of the floxed *Mmp14* allele and the absence of MT1-MMP mRNA (Fig EV1A–C). MT1-MMP expression was also reduced in the colonic capillaries from MT1$^{iΔEC}$ mice examined at 7 days post-DSS (Fig EV1D).

To test the effect of prophylactic endothelial MT1-MMP deletion on DSS-induced IA, we implemented a 3D imaging method based on high-resolution confocal microscopy and Imaris® image analysis of whole-mount CD31-stained mouse intestine; this method allowed us to quantify holes in the vascular tri-corners indicating intraluminal pillars as well as capillary loops and duplications in the mucosal plexus, all hallmarks of IA (Fig EV2A). We also measured capillary bifurcation angles in the mucosal plexus, which decrease during IA (Fig EV2B; Ackermann *et al*, 2013). DSS-induced mild colitis in MT1$^{f/f}$;*Cdh5*Cre$^{ERT2}$-negative (MT1$^{f/f}$) control mice was associated with significantly increased IA events (holes, loops, and duplications) and smaller capillary bifurcation angles at 3 days compared

with non-treated mice (Fig 1A–C). Of note, MT1$^{i\Delta EC}$ mice showed significantly reduced numbers of DSS-induced IA capillary events (holes, loops, and duplications) at 3 and 7 days compared with MT1$^{f/f}$ mice (Fig 1A and B); intercapillary angles were also preserved in the colon mucosal plexus of MT1$^{i\Delta EC}$ mice at 3 days post-DSS treatment (Fig 1C). These findings identify MT1-MMP as a novel endothelial actor in inflammatory IA.

### Loss of endothelial MT1-MMP preserves intestinal vascular perfusion and ameliorates colitis

To investigate the impact of inflammatory IA on vascular function, we analyzed vessel perfusion by intravascular injection of isolectin B4 (IB4). Perfusion decreased during colitis progression in the mucosa vascular plexus of both MT1$^{f/f}$ and MT1$^{i\Delta EC}$ mice, but it was significantly better preserved in the latter at 7 days post-1%

DSS treatment (Fig 2A and B). Furthermore, 1% DSS induced vascular leakage at days 3 and 7 in the colonic mucosa of MT1$^{f/f}$ mice and at a lower extent in MT1$^{i\Delta EC}$ mice (Appendix Fig S2A).

Treatment with 1% DSS produced only subtle alterations in the intestinal mucosa after 3 days, but at 7 days, hematoxylin and eosin histology revealed better preservation of colon morphology in MT1$^{i\Delta EC}$ mice, showing fewer areas of crypt destruction than control mice (Fig 2C). Second-harmonic generation (SHG) microscopy confirmed the presence of well-structured collagen fibers surrounding crypts in MT1$^{i\Delta EC}$ mice at 7 days post-DSS, contrasting with abundant and disorganized collagen fibers in control mice (Fig 2D), an additional sign of enhanced tissue damage and fibrosis. To estimate disease severity over the 7 days of 1% DSS treatment, we calculated the disease activity index (DAI), based on a composite of weight loss, stool consistency, and hemorrhage (see Methods). The DAI was significantly lower in MT1$^{i\Delta EC}$ mice than in MT1$^{f/f}$ controls

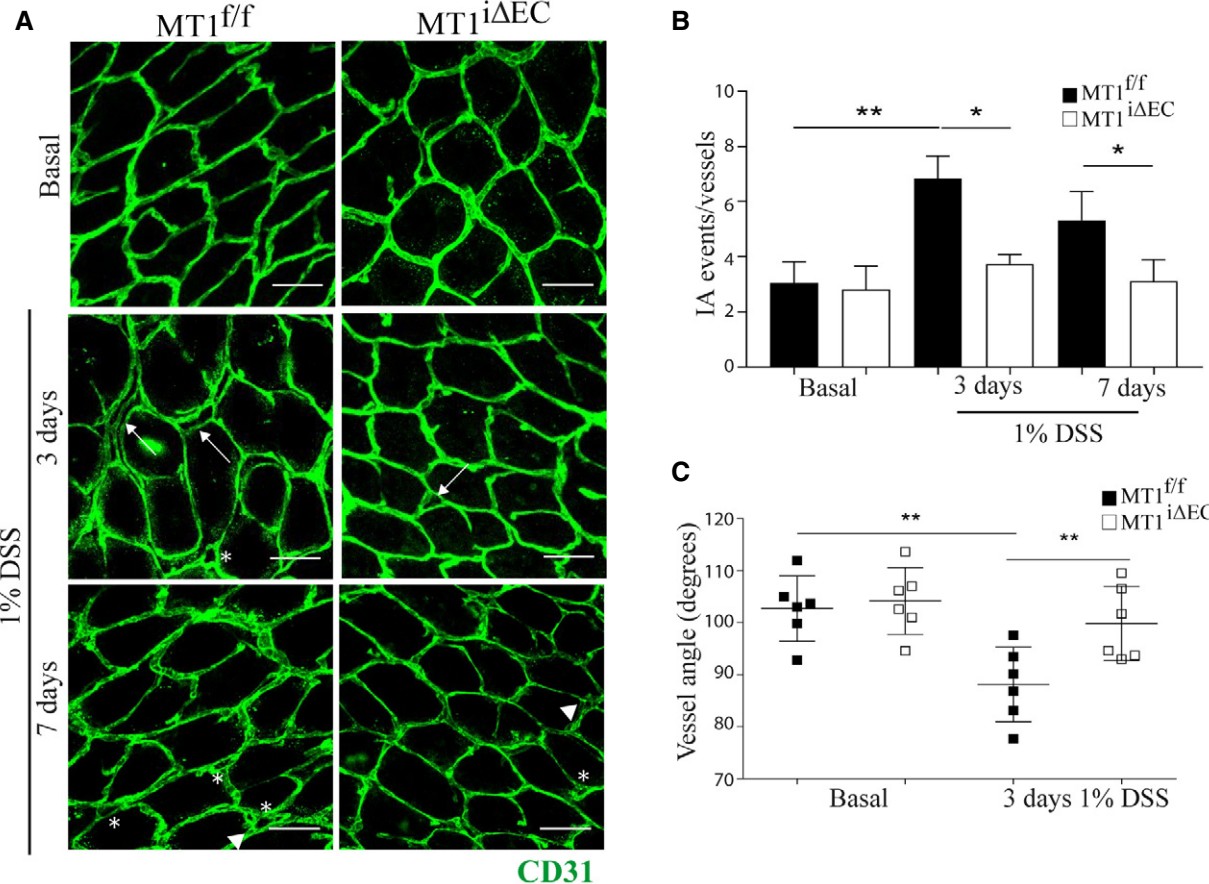

**Figure 1. MT1-MMP expression in endothelial cells contributes to intussusceptive angiogenesis in colitis.**

A  Representative maximum-intensity projection images of colon mucosal plexus in CD31-stained (green) whole-mount distal colon from MT1$^{f/f}$ and MT1$^{i\Delta EC}$ mice left untreated (basal) or treated with 1% DSS for 3 or 7 days. Scale bar, 40 μm. Arrows, arrowheads, and asterisks indicate duplications, loops, and pillars, respectively.

B  Quantification of IA events in the colon mucosal plexus of mice treated as in (A), including vascular holes, duplications, and loops; $n$ = 9–15 mice per genotype and condition. Data are shown as mean ± SEM and were tested by one-way ANOVA with Benjamini and Hochberg post-test; *$P < 0.05$, **$P < 0.01$.

C  Quantification of vascular angles at the Y-junctions in the colon mucosal plexus of mice left untreated or treated with 1% DSS for 3 days; $n$ = 6 mice per genotype and condition. Data are shown as mean ± SEM and additionally as individual animal values and were tested by one-way ANOVA with Benjamini and Hochberg post-test; **$P < 0.01$.

Data information: Please see Appendix Table S3 for exact $P$-values.
Source data are available online for this figure.

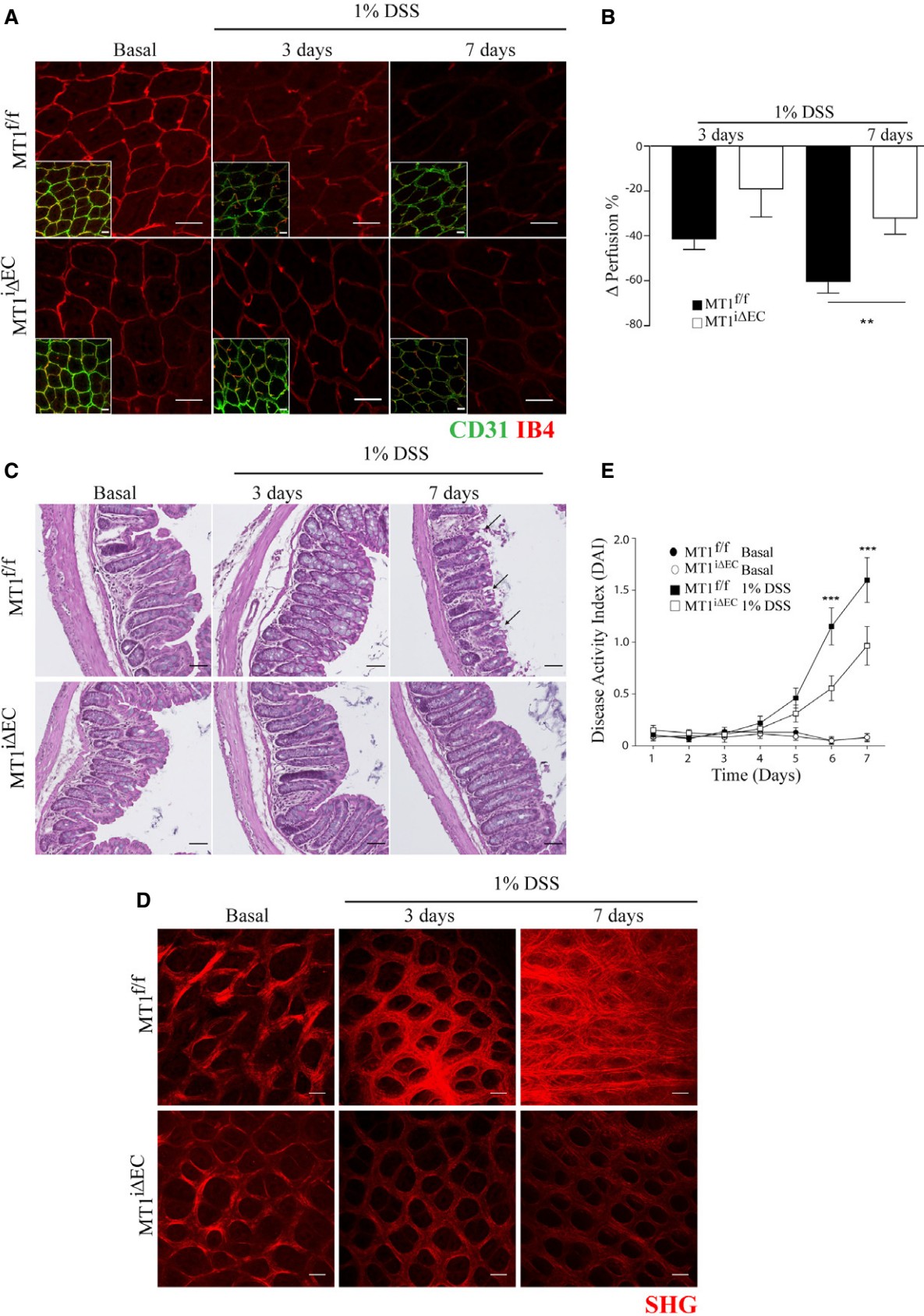

Figure 2.

**Figure 2. MT1-MMP absence from endothelial cells limits deterioration of vascular perfusion and impedes colitis progression.**

A   Representative maximum-intensity projection images of whole-mount distal colons stained for CD31 (green) and IB4 (red) in MT1[f/f] and MT1[iΔEC] mice left untreated (basal) or treated with 1% DSS for 3 or 7 days. Scale bar, 40 μm.

B   Perfusion decreased during 1% DSS-induced colitis in MT1[f/f] and MT1[iΔEC] mice; $n = 8–12$ mice per genotype and condition. Data are shown as mean ± SEM and were tested by one-way ANOVA with Benjamini and Hochberg post-test; $**P < 0.01$.

C   Representative H&E-stained colon sections from MT1[f/f] and MT1[iΔEC] mice left untreated (basal) or treated with 1% DSS for 3 or 7 days. Arrows indicate crypt destruction. Scale bar, 50 μm.

D   Representative second-harmonic generation (SHG) microscopy images of mucosal plexus in whole-mount colons from MT1[f/f] or MT1[iΔEC] mice left untreated (basal) or treated with 1% DSS for 3 or 7 days. Scale bar, 40 μm.

E   Disease activity index (DAI, a composite of weight change, stool consistency, and presence of fecal blood) during 1% DSS-induced colitis in MT1[f/f] and MT1[iΔEC] mice. Untreated mice were included as a control. $n = 19–24$ per genotype and condition. Data are shown as mean ± SEM and were tested by two-way ANOVA with Benjamini and Hochberg post-test; $***P < 0.001$.

Data information: Please see Appendix Table S3 for exact *P*-values.
Source data are available online for this figure.

from DSS day 5 onward, indicating milder disease in the absence of endothelial MT1-MMP (Fig 2E). Numbers of CD11b[+] leukocytes (mostly neutrophils and monocytes) did not differ between genotypes at DSS day 3, when IA was already decreased in MT1[iΔEC] mice; in contrast, at DSS day 7, CD11b[+] leukocyte numbers in MT1[iΔEC] mice showed a slight but significant reduction compared with MT1[f/f] controls (Appendix Fig S2B and C). Thus, endothelium-specific MT1-MMP deletion improves 1% DSS-induced colitis progression primarily through an early impact on IA events at 3 days rather than on vascular perfusion, leukocyte traffic, and tissue alterations modulated at 7 days.

In light of these results, we next tested the therapeutic action of endothelial MT1-MMP deletion on colitis progression. Mice received five daily 4-OHT injections beginning 4 days after the initiation of 1% DSS treatment, and mice were sacrificed at DSS day 15. Therapeutic endothelial MT1-MMP deletion during established colitis significantly reduced weight loss and DAI from day 9 onward compared with MT1[f/f] control mice (Fig EV3).

## MT1-MMP expression in ECs is required for vasodilation and NO production

The cellular and molecular mechanisms underlying IA are poorly defined, but there is a general consensus that it is initiated by changes in hemodynamic forces and increased blood flow (Filipovic *et al*, 2009). To investigate the role of endothelial MT1-MMP in this early step of IA that involves vasodilation (Fig 3A), we relied on intravital microscopy of the cremaster muscle and monitored arteriole vasodilation by injecting acetylcholine (ACh), which triggers EC NO secretion (Rius & Sanz, 2015; Fig 3B–D). No differences between genotypes were observed in arteriole diameter (~ 30 μm) at baseline (Fig 3C, and Movies EV1 and EV2); in contrast, maximal ACh-induced vasodilation (reached 3 min after injection) was significantly impaired in arterioles of MT1[iΔEC] mice compared with MT1[f/f] controls (Fig 3D, and Movies EV1 and EV2). Given the observation of IA events in colon mucosal plexus capillaries, we also analyzed the capillaries in the cremaster muscle. The diameter of cremaster capillaries did not increase in MT1[iΔEC] mice after Ach injection; however, this impaired response was rescued by i.v. injection of the NO donor DEANO (Fig 3E and F). These results suggest that impaired vasodilation in MT1-MMP-null ECs may be due to decreased NO production. Low NO production was confirmed by DAF-FM analysis of aortic endothelial cells (MAEC) isolated from

MT1[iΔEC] mice (Appendix Fig S3). Furthermore, human umbilical vein endothelial cells (HUVEC) expressing MT1-MMP-targeting siRNA (siMT1) not only produced significantly lower amounts of NO (Fig 4A) but also expressed lower amounts of eNOS protein and eNOS (*Nos3*) mRNA than corresponding control ECs (Fig 4B–D).

We next implemented a protocol to assess VEGF-induced vasodilation of the colon mucosal plexus by its administration via a rectal cannula to get closer to the pathophysiology of IA during colitis (Fig EV4A and B). As shown in Fig EV4C and D, the diameter of the mucosa plexus capillaries was significantly reduced in VEGF-treated MT1[iΔEC] mice compared to MT1[f/f] controls.

## MT1-MMP catalytic activity is required for NO production and colitis-induced IA

To determine whether inflammatory IA regulation by MT1-MMP depends on its catalytic or signaling activities (Gonzalo *et al*, 2010; Koziol *et al*, 2012a), MT1[f/f];*Cdh5*Cre[ERT2] mice were inoculated with lentivirus (LV) encoding full-length (FL) MT1-MMP or mutated versions that abolish its catalytic (E240A) or signaling (Y573F) activities. Daily 4-OHT injections to delete endogenous endothelial MT1-MMP were started 72 h after LV injection, and 1% DSS treatment was initiated 3 days later (Fig 5A and B). Western blot analysis of GFP expression confirmed similar transduction levels in the colon regardless of the injected LV (Fig 5C). LV encoding FL and Y573F MT1-MMP increased IA events 3 days post-DSS to levels comparable to those observed in LV-mock-injected MT1[f/f] mice, whereas LV encoding E240A MT1-MMP had no effect and injected mice showed similar numbers of IA events than LV-mock-injected MT1[iΔEC] mice (Fig 5D–F). These results demonstrate that IA induced by 1% DSS colitis requires the catalytic activity of MT1-MMP.

## MT1-MMP cleavage of TSP1 drives EC NO production

We next tested the effect of the inhibitory antibody anti-MT1-MMP LEM-2/15 (Galvez *et al*, 2001) on NO production. The inhibitory LEM-2/15 antibody significantly reduced NO amount in HUVEC transfected with control siRNA (siControl) in comparison with non-treated or isotype control-treated cells, whereas the antibody had no effect on NO production by siMT1-silenced HUVEC (Appendix Fig S4A). MT1-MMP catalytic activity is thus also required for EC NO production, which likely acts upstream of inflammatory IA (Williams *et al*, 2006).

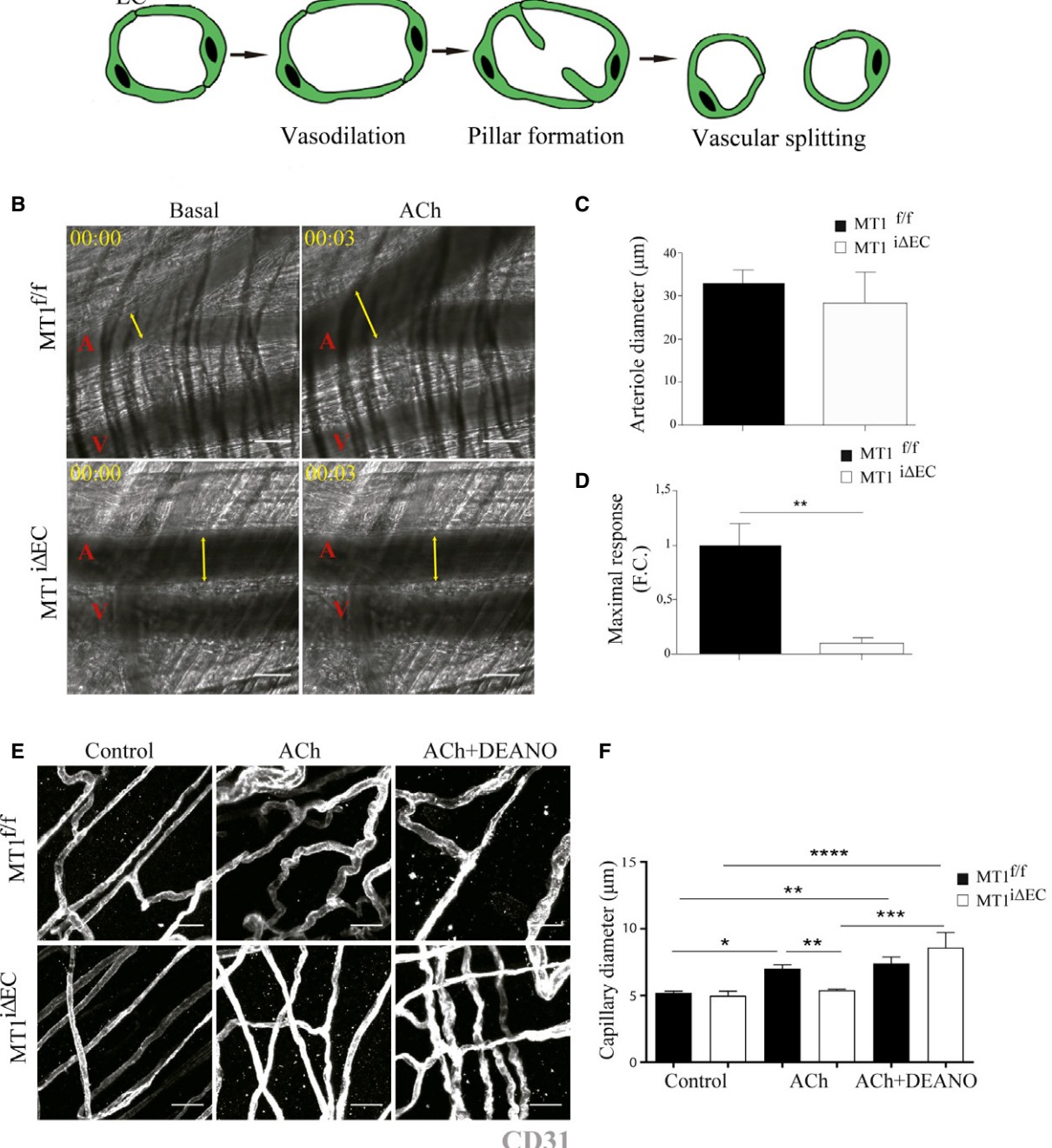

**Figure 3. Endothelial cell MT1-MMP expression is required for nitric oxide-dependent vasodilation *in vivo*.**

A    Scheme showing IA process including vasodilatation, pillar formation, and vascular splitting. EC, endothelial cell.

B    Representative intravital microscopy images of cremaster muscle from MT1^f/f or MT1^iΔEC mice before and 15 min after i.v. injection of 100 μl of 100 μM acetylcholine (ACh). A and V mark arterioles and venules, respectively, and yellow arrows indicate arteriole diameter. Elapsed time is designated as hh:mm. Scale bar, 30 μm.

C, D    Cremaster arteriole diameter in basal conditions (C) and the maximum vasodilation response (diameter fold change) after 3 min of ACh stimulation (D); n = 5–6 mice per genotype. Data are shown as mean ± SEM and were tested by paired *t*-test; **P < 0.01.

E    Representative images of staining for CD31 (gray) in whole-mount cremaster muscle dissected from MT1^f/f or MT1^iΔEC mice before and 15 min after i.v. injection of 100 μl of 100 μM ACh alone or in combination with DEANO (100 μl of 10^−4 M). Scale bar, 25 μm.

F    Quantification of capillary diameter in mice analyzed as in (E); n = 5–8 mice per genotype and condition. Data are shown as mean ± SEM and were tested by one-way ANOVA with Benjamini and Hochberg post-test; *P < 0.05, **P < 0.01, ***P < 0.001, ****P < 0.0001.

Data information: Please see Appendix Table S3 for exact *P*-values.
Source data are available online for this figure.

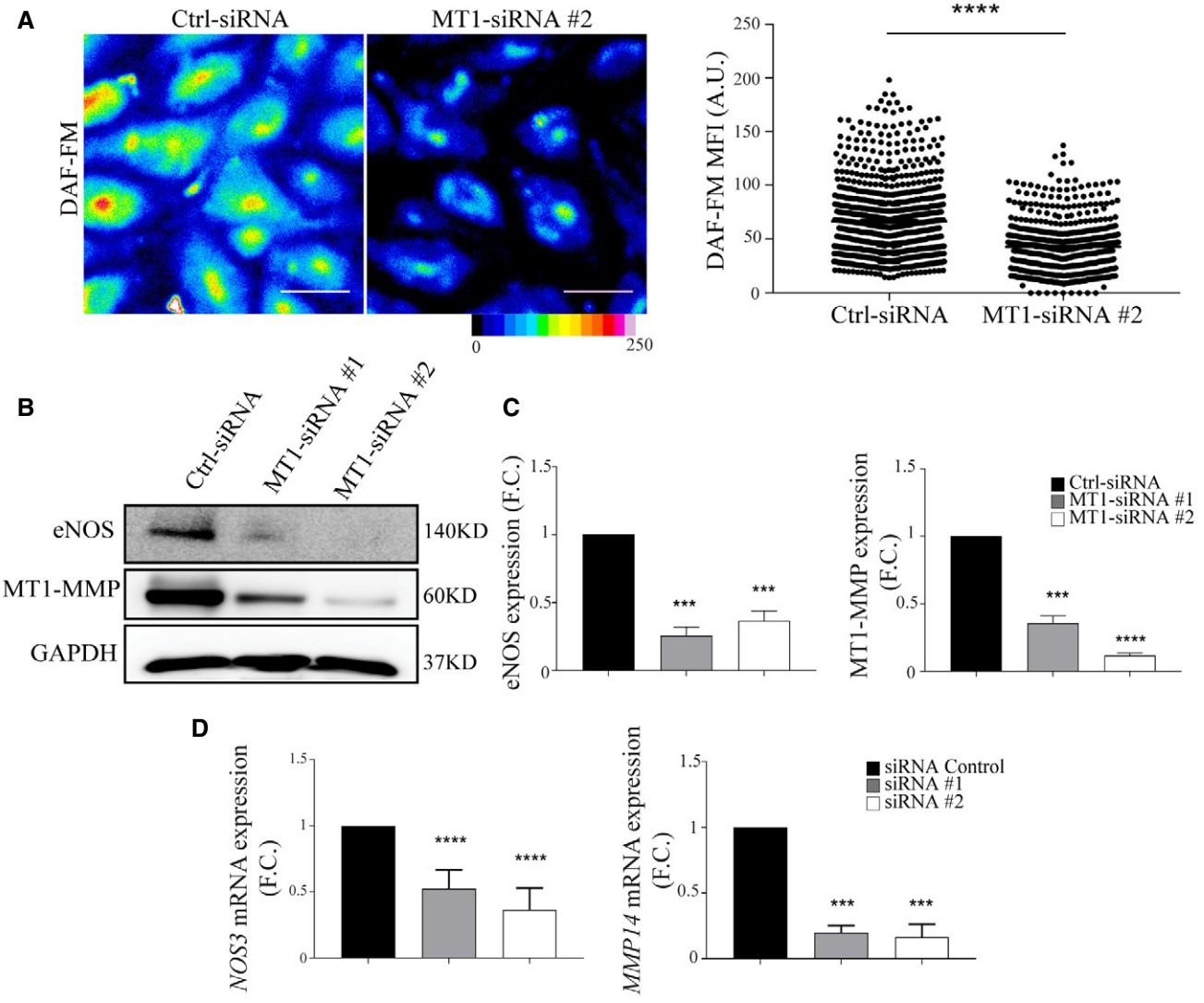

**Figure 4. Decreased nitric oxide production and eNOS expression in MT1-MMP-silenced human endothelial cells.**

A    Representative pseudo-colored microscopy images of DAF-FM (4-amino-5-methylamino-2′,7′-difluorofluorescein diacetate) fluorescence intensity in human endothelial cells (HUVEC) expressing control or MT1-MMP siRNA (left); scale bar, 50 μm. The graph to the right shows DAF-FM mean fluorescence intensity (MFI) values obtained in $n$ = 637–879 cells analyzed per condition in six independent experiments. Data are shown as individual cell values and mean ± SEM and were tested by the Mann–Whitney test; ****$P$ < 0.0001.

B, C  Representative Western blot of eNOS and MT1-MMP in HUVEC expressing control or different MT1-MMP siRNAs; GAPDH is included as a loading control (B). Bar graphs (C) show relative eNOS and MT1-MMP expression (vs. GAPDH); $n$ = 5 independent experiments. Data are shown as mean ± SEM and were tested by one-sample $t$-test; ***$P$ < 0.001, ****$P$ < 0.0001.

D    qPCR analysis of relative eNOS (*NOS3*) and MT1-MMP (*MMP14*) mRNA levels in HUVEC expressing control or MT1-MMP siRNAs; $n$ = 4 independent experiments. F.C., fold change. Data are shown as mean ± SEM and were tested by one-sample $t$-test; ***$P$ < 0.001, ****$P$ < 0.0001.

Data information: Please see Appendix Table S3 for exact $P$-values.
Source data are available online for this figure.

Several angiogenesis pathways, including mechanisms involving eNOS, are suppressed by the matricellular protein thrombospondin-1 (TSP1) through its binding to CD36 (nM affinity) and/or CD47/IAP (pM affinity) (Lawler & Lawler, 2012; Resovi et al, 2014). We had previously identified TSP1 as a master MT1-MMP substrate and showed that loss of MT1-MMP reduced TSP1 release in TNFα-stimulated ECs (Koziol et al, 2012a). In the context of colitis, TSP1 serum levels increased after 7 days of 1% DSS treatment

(20.27 ± 11.84 pg/ml in control vs. 37.04 ± 1.21 pg/ml in treated mice) and we sought to analyze the impact of endothelial MT1-MMP in such release. For that, we examined TSP1 intestinal distribution in DSS-treated mice, using an antibody against a sequence near the TSP1 N-terminus (Lee et al, 2006). Immunostained sections revealed that 1% DSS treatment upregulated TSP1 expression in the mouse colon after 3 days (Fig 6A and B). Moreover, image analysis demonstrated that TSP1 was significantly accumulated forming a

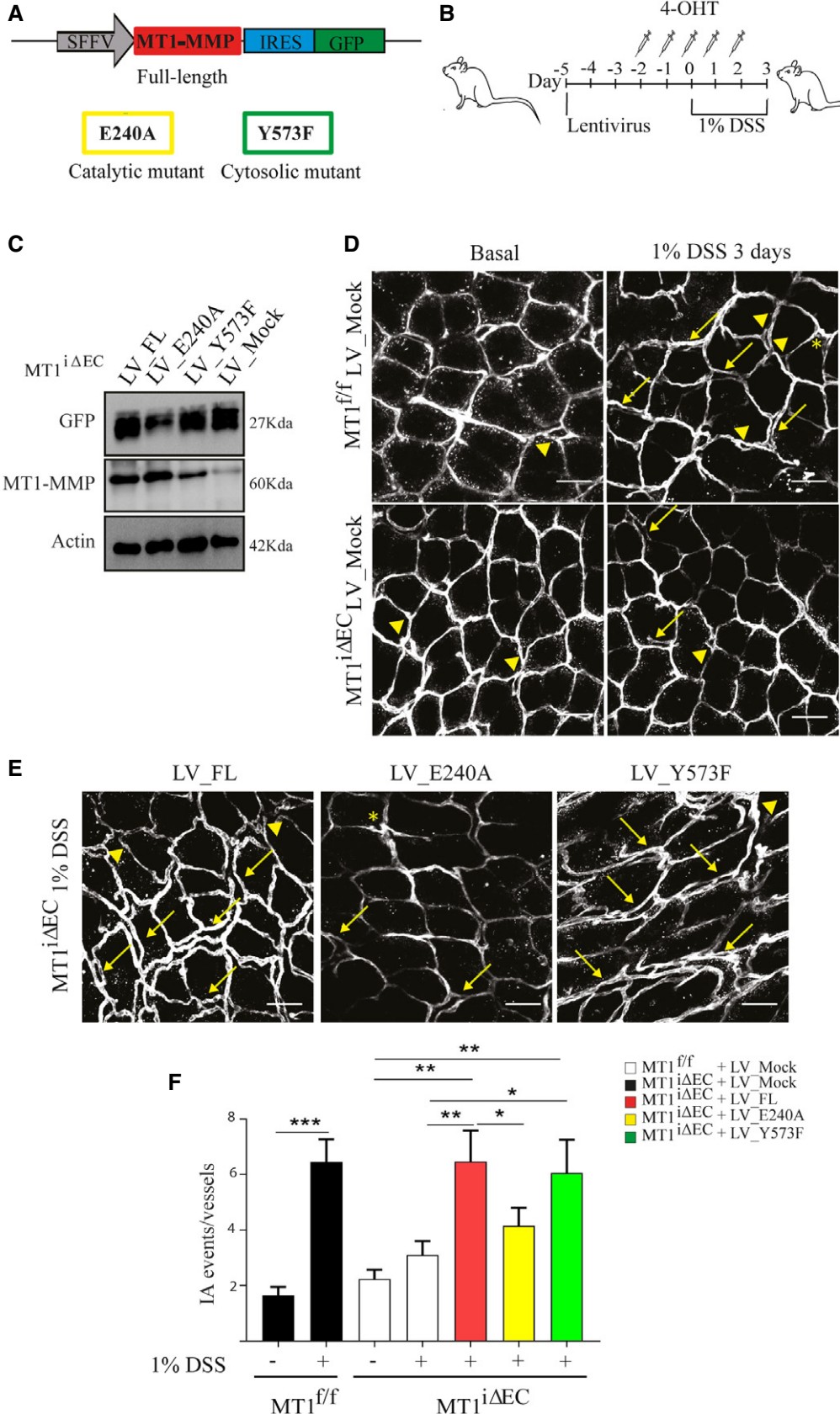

Figure 5.

**Figure 5.   MT1-MMP actions in intussusceptive angiogenesis during colitis depend on its catalytic activity.**

- A    Lentivirus (LV) construct used to drive expression of MT1-MMP (*MMP14*) and green fluorescent protein (GFP) under the SFFV and IRES promoters, respectively. The point mutations introduced to obtain the catalytic (E240A) and signaling (Y573F) mutants are indicated.
- B    Experimental design for i.v. lentiviral injection in mice and induction of mild colitis with 1% DSS.
- C    Western blot of GFP and MT1-MMP protein expression in colon lysates from MT1^iΔEC mice previously injected with mock LV or LV encoding full-length MT1-MMP or the E240A or Y573F mutants; mice were sacrificed 3 days after 1% DSS treatment. Actin is included as a loading control.
- D, E   Representative maximum-intensity projection images of staining for CD31 (gray) in whole-mount colons from MT1^f/f or MT1^iΔEC mice previously injected with mock LV (D) or LV encoding full-length MT1-MMP or the E240A or Y573F mutants (E); mice were sacrificed 3 days after 1% DSS treatment. Scale bar, 40 μm. Arrows, arrowheads, and asterisks indicate duplications, loops, and pillars, respectively.
- F    Quantification of IA events in mice treated as in (D and E); *n* = 5–8 mice per genotype and condition. Data are shown as mean ± SEM and were tested by one-way ANOVA with Benjamini and Hochberg post-test; *P < 0.05, **P < 0.01, ***P < 0.001.

Data information: Please see Appendix Table S3 for exact *P*-values.
Source data are available online for this figure.

thick cuff around MT1-MMP-negative vessels of the colon mucosa nearby the muscularis mucosa in MT1^iΔEC mice compared with less TSP1 abundance around the MT1-MMP-expressing corresponding vessels in MT1^f/f mice (Fig 6B and C). These data indicate that endothelial MT1-MMP contributes to perivascular TSP1 processing in the intestine.

TSP1 also accumulated in siMT1-silenced HUVEC (Fig 7A), related to defective TSP1 processing in the absence of the protease (Koziol *et al*, 2012a). Moreover, culture on TSP1-coated plates increased NO production in siControl-HUVEC significantly more than in siMT1-silenced HUVEC, suggesting that signals mediated by processed TSP1 contribute to increased NO production in MT1-MMP-expressing ECs (Appendix Fig S4B). Accordingly, we detected an N-terminal TSP1 fragment of about 50 KD in lysates from MT1-MMP-expressing HUVEC whose abundance was reduced in MT1-MMP-silenced cells, and we confirmed that TSP1 cleavage by the MT1-MMP catalytic domain yielded that N-terminal fragment by *in vitro* digestion assays (Fig EV5A and B). To decipher how MT1-MMP-mediated TSP1 processing affects NO production, we combined a CleavPredict search (Kumar *et al*, 2015) with *in silico* protein modeling. This approach identified positions H$^{441}$W and P$^{467}$Q in TSP1 as having good accessibility and proximity to the protease catalytic pocket, suggesting them as candidate sites for MT1-MMP cleavage (Appendix Table S1, and Fig EV5C and D), and consistent with the N-terminal TSP1 fragment observed in HUVEC lysates and by *in vitro* digestion (Fig EV5A and B). Moreover, the amino acids flanking the predicted cleavage sites (P1′:P1 positions) had high MEROPS database scores (4:1 and 8:6 for H$^{441}$W and P$^{467}$Q sites, respectively; https://www.ebi.ac.uk/merops/). Cleavage of TSP1 by MT1-MMP at H$^{441}$W and/or P$^{467}$Q would likely disrupt the CD36-binding motifs and generate a C-terminal fragment with preserved binding sites for CD47/IAP and its partner αvβ3 integrin (Lindberg *et al*, 1996), which would mediate MT1-MMP-dependent NO production in ECs. To directly validate this hypothesis, we investigated the effect of the E123CaG-1 human TSP1 fragment (aa 549–1,170; Margosio *et al*, 2008) spanning most of the predicted C-terminal TSP1 fragment after MT1-MMP processing (441/467–1,170) and containing CD47 and αvβ3 integrin binding sites. We confirmed that the E123CaG-1 TSP1 fragment induced NO production in MT1-MMP-deficient HUVEC compared with full-length TSP1 (Fig EV5E).

Interaction between CD47 and TSP1 influences NO signaling (Rogers *et al*, 2014), and we found that an anti-CD47 antibody known to block CD47 interaction with the C-terminal TSP1 domain

reduced NO production by MT1-expressing ECs but had no further effect on NO production by siMT1-silenced HUVEC (Fig 7B). To determine whether CD47-driven effects involve the activity of its partner αvβ3 integrin, we blocked αvβ3 integrin binding to TSP1 with an inhibitory anti-αv integrin antibody, an RGDS peptide, or cilengitide (based on the cyclic peptide cyclo-RGDfV). All three strategies significantly decreased NO production by MT1-expressing ECs; in contrast, these treatments had negligible impact on the already reduced NO production by siMT1-silenced cells (Fig 7B–D). Similarly, NO production by siControl- and siMT1-transduced cells was unaffected by the control peptide RADS or isotype control IgG. The RGDS peptide can compete for αvβ3 integrin binding to several matrix proteins; therefore, to test for a specific effect via TSP1, we tested the effect on NO production by ECs of a selective nonamer spanning the RGD binding site for αvβ3 integrin in TSP1. The nonamer GDGRGDACK, but not the control peptide GDGRADACK, reduced NO production in MT1-MMP-expressing endothelial cells but not in siMT1-silenced cells (Fig 7E).

## MT1-MMP/TSP1 signaling provides a possible diagnostic and therapeutic target in IBD

To explore the possible implications of endothelial MT1-MMP-mediated TSP1 processing in patients with IBD, we first checked MT1-MMP and TSP1 vascular expression and distribution in unaffected and affected colon areas of patients suffering from ulcerative colitis or Crohn's disease. Immunostaining on colon sections revealed more abundant capillaries in affected vs. unaffected areas that were accompanied by MT1-MMP and TSP1 upregulated expression and distinct tissue distribution; MT1-MMP was mainly located in CD31-positive vessels of the inflamed intestine while TSP1 was sparse around these vessels and mostly expressed by spiky cells resembling macrophages (Fig 8A). Since TSP1 was stained with the monoclonal antibody A6.1, which recognizes an epitope downstream of the predicted TSP1 cleavage sites for MT1-MMP (Annis *et al*, 2006), this pattern is consistent with TSP1 cleavage by endothelial MT1-MMP during DSS-induced colitis leading to TSP1 paucity around MT1-MMP-expressing vessels in MT1^f/f mice (Fig 6B and C) and to its release to the serum. We then quantified serum TSP1 in a cohort of IBD patients scored for clinical activity, including ≥ 50 patients with ulcerative colitis or with Crohn's disease. Remarkably, TSP1 was significantly more abundant in the serum from patients with mildly active (but not with highly active) ulcerative colitis or Crohn's disease compared to healthy controls

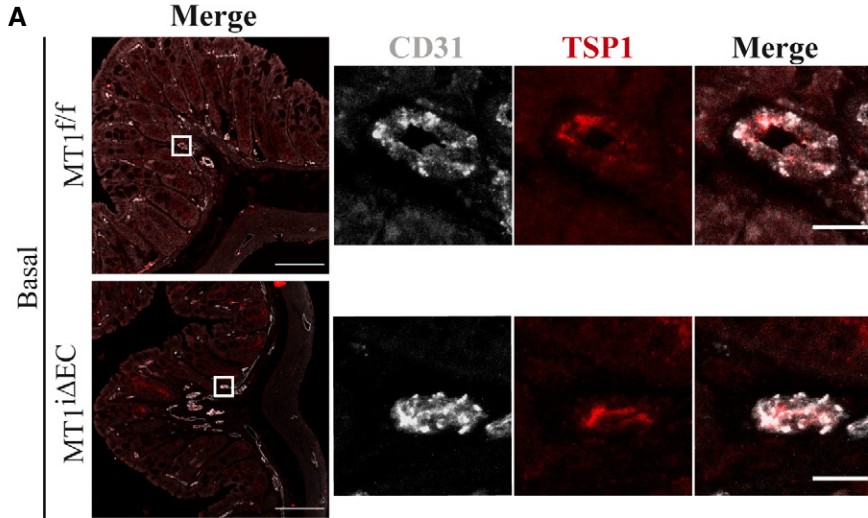

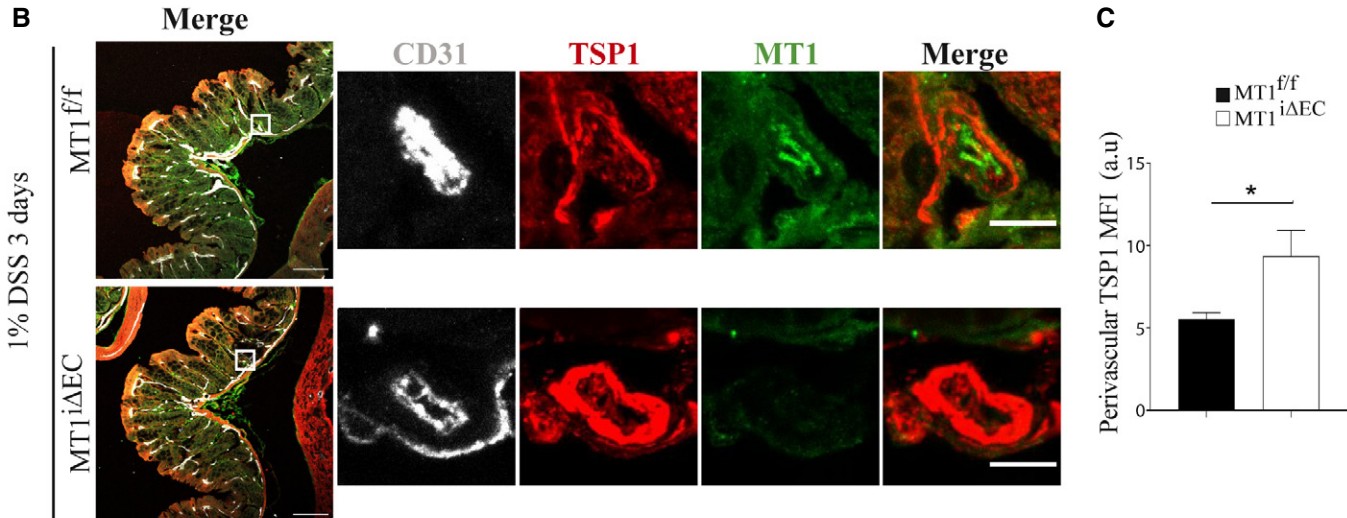

**Figure 6. MT1-MMP absence in endothelial cells leads to perivascular accumulation of thrombospondin-1 (TSP1) in the inflamed colon.**

A, B Representative maximum-intensity projection images of staining for TSP1 (red) and CD31 (gray) in (A) and also for MT1-MMP (green) in (B) in colon sections obtained from MT1$^{f/f}$ and MT1$^{i\Delta EC}$ mice left untreated (A) or treated with 1% DSS for 3 days (B). Magnified views are shown to the right. Scale bars, 100 μm in main panels and 10 μm in the magnified views.

C TSP1 mean fluorescence intensity (MFI) in the perivascular area of large vessels present in the colon mucosa of treated mice, as depicted in (B); n = 4 mice per genotype. Data are shown as mean ± SEM and were tested by t-test; *P < 0.05.

Data information: Please see Appendix Table S3 for exact P-values.
Source data are available online for this figure.

(Fig 8B). This finding thus suggests that serum TSP1 may be a biomarker of early or mild IBD.

To evaluate the relevance of the MT1-MMP/TSP1/nitric oxide pathway in colitis *in vivo*, we first injected intraperitoneally the C-terminal E123CaG-1 TSP1 fragment (containing CD47 and αvβ3 integrin binding sites) into MT1$^{i\Delta EC}$ mice and observed that it significantly increased the number of IA events 3 days post-1% DSS

treatment compared with mice treated with full-length TSP1 (Fig EV5F and G). Contrarily, blocking the generation of the TSP1 fragment by intraperitoneal injection of the anti-MT1-MMP inhibitory antibody LEM-2/15 significantly decreased the number of IA events in the mucosal plexus and conserved well-organized collagen fibers in MT1$^{f/f}$ control mice after 3 days of 1% DSS treatment compared to isotype control-treated mice (Fig 8C–E). Given our

finding that αvβ3 integrin binding to TSP1 drives NO production in ECs *in vitro*, we finally investigated the therapeutic potential of this pathway by selectively blocking this interaction *in vivo* and

checking the impact on IA and colitis. MT1$^{f/f}$ control mice were fitted with subcutaneous minipumps allowing continuous release of a high dose of the TSP1 nonamer GDGRGDACK or the control

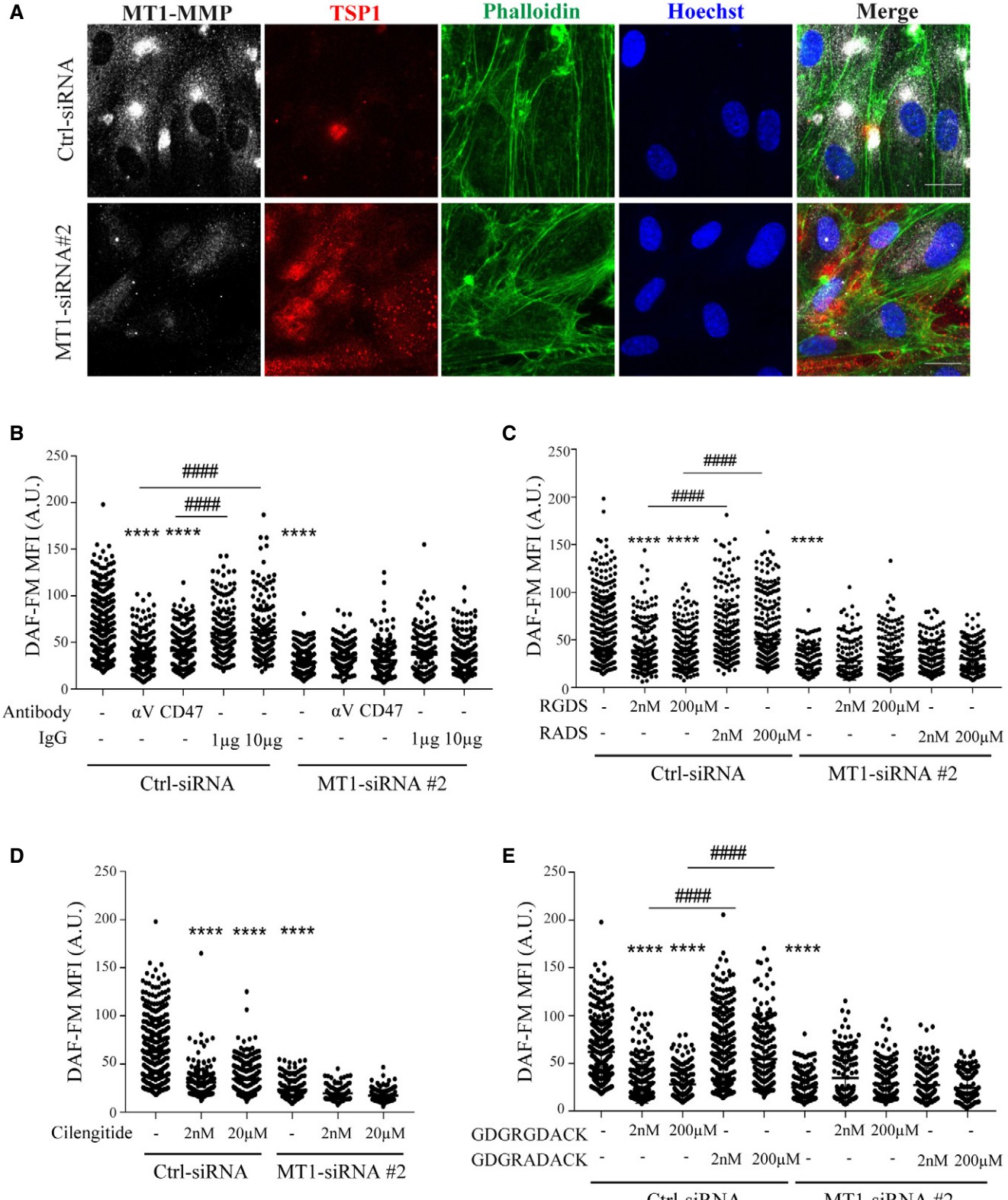

**Figure 7.**

◄

**Figure 7. Binding of TSP1 to CD47/IAP and αvβ3 integrin regulates nitric oxide production in MT1-MMP-expressing endothelial cells.**

A   Representative maximum-intensity projection images of HUVEC expressing control or MT1-MMP siRNA and stained for MT1-MMP (gray), TSP1 (red), and F-actin (phalloidin, green); nuclei are stained with Hoechst (blue). Scale bar, 20 μm.

B   DAF-FM mean fluorescence intensity (MFI) in HUVEC expressing control or MT1-MMP siRNA and left untreated or treated with blocking anti-CD47 or anti-αv integrin antibodies (1 and 10 μg/ml, respectively, for 24 h) or their corresponding IgG isotype controls (IgG); n = 149–360 cells analyzed per condition in three independent experiments.

C   DAF-FM mean fluorescence intensity (MFI) in HUVEC expressing control or MT1-MMP siRNA and left untreated or treated with 2 nM or 200 μM RGDS or its control peptide RADS; n = 147–385 cells analyzed per condition in three independent experiments.

D   DAF-FM mean fluorescence intensity (MFI) in HUVEC expressing control or MT1-MMP siRNA and left untreated or treated with 2 nM or 20 μM of the RGD cyclic peptide cilengitide; n = 100–360 cells analyzed per condition in three independent experiments.

E   DAF-FM mean fluorescence intensity (MFI) in HUVEC expressing control or MT1-MMP siRNA and left untreated or treated with 2 nM or 200 μM of the nonamer GDGRGDACK or its control peptide GDGRADACK; n = 149–337 cells analyzed per condition in three independent experiments.

Data information: Data are shown in all panels as individual cell values and mean ± SEM and were tested by the Kruskal–Wallis test; ****$P$ < 0.0001, ####$P$ < 0.0001.
* indicates comparison with Ctrl-siRNA, and # indicates comparison of the condition with its corresponding control. Please see Appendix Table S3 for exact $P$-values.
Source data are available online for this figure.

GDGRADACK (equivalent dose of 2.4 mg/mouse/day). Mice were then administered 1% DSS for 3 days, and colonic IA events were analyzed after sacrifice. Mice treated with the TSP1-nonamer GDGRGDACK had significantly fewer IA events than mice treated with the control peptide (Fig 8F and G). The reduced IA in GDGRGDACK-treated mice also resulted in better-preserved collagen fiber organization assessed by second-SHG microscopy (Fig 8H).

## Discussion

In this study, we identify the protease MT1-MMP as an endothelial actor in IA, particularly during the pathogenesis of inflammatory colitis. Our results point to NO production via an MT1-MMP/TSP1/integrin αvβ3/eNOS pathway as the MT1-MMP catalytic-dependent mechanism underlying vasodilation during this mode of angiogenesis (Fig 9). Evidence from human samples confirms that this pathway is active in IBD, and blockade of DSS-induced colitis in mice with an MT1-MMP inhibitory antibody or a TSP1-targeting peptide indicates the potential of this strategy to improve IBD.

Our work also establishes that confocal microscopy of vessel-stained whole-mount colons combined with 3D-image rendering provides sufficient resolution to identify and quantify capillary holes/pillars, loops, and duplications, all hallmarks of IA, circumventing the limitations of other techniques (Nowak-Sliwinska et al, 2018). The use of this imaging approach in fluorescent reporter mouse lines may allow the detection of intraluminal pillars and determination of the contribution of other cell types to pathophysiological IA.

The EC-autonomous functions of MT1-MMP in vivo have remained poorly defined due to the unavailability, until recently, of conditionally deficient mice (Klose et al, 2013; Tang et al, 2013; Gutierrez-Fernandez et al, 2015; Rafii et al, 2015). Our results demonstrate that the specific absence of MT1-MMP in ECs decreases the number of IA events early during 1% DSS-induced mild colitis in mice and ameliorates the disease. These early IA events occur when only a minor inflammatory infiltrate is present, suggesting that the 1% DSS mild colitis model may be especially suited to dissecting the vascular IA response independently of inflammation (Chidlow et al, 2006). Our data also indicate that MT1-MMP endothelial targeting could be useful in other conditions that involve pathological IA, such as cancer (Ribatti & Djonov, 2012) and also

bronchopulmonary dysplasia of the pre-term lung, in light of the reported role of MT1-MMP in postnatal lung expansion (De Paepe et al, 2017; Oblander et al, 2005).

Regarding the mechanism by which MT1-MMP contributes to IA and colitis, our results show that MT1-MMP EC deficiency results in decreased NO-dependent vasodilation of cremaster muscle arterioles and capillaries in vivo and establish that MT1-MMP catalytic activity is required for NO production by ECs in vitro, regulating both eNOS protein and mRNA levels. Our data therefore confirm a pathogenic role for endothelial NO in colitis in line with Beck et al (Beck et al, 2004) and contrasting its reported healing and protection actions, which are likely related to an impact on cell types other than the endothelium (Sasaki et al, 2003; Aoi et al, 2008). The dependence on endothelial MT1-MMP of colon mucosa capillary vasodilation induced by VEGF, a driver of IA and recognized activator or nitric oxide production (Cooke & Losordo, 2002), underscores the relevance of this pathway in colitis pathophysiology.

MT1-MMP may play a dual, bimodal role via NO production during IA initiation and progression. MT1-MMP-driven NO production will primarily increase blood flow in the arterioles and therefore in the downstream capillary plexus, promoting vessel splitting. This is supported by the predominant constitutive expression of MT1-MMP in arteriolar ECs rather than in the colonic mucosa capillaries. But MT1-MMP is upregulated after DSS treatment in the ECs of the mucosal plexus particularly near the Y-junctions where IA events often occur. At these sites, blood flow gradients and decreased shear stress lead to intraluminal pillar formation by undefined mechanisms (Filipovic et al, 2009). MT1-MMP may also drive NO production at these sites, which could then contribute to intraluminal EC rearrangements and pillar formation. Accordingly, lack of NO actions in the skeletal muscle led to reduced numbers of intraluminal filopodia, ultimately hampering cell rearrangements needed for capillary splitting (Williams et al, 2006). NO is also induced and required during shear stress-driven angiogenesis (Kolluru et al, 2010) and can regulate EC junction remodeling and directional migration (Noiri et al, 1998; Kevil et al, 2004; Thibeault et al, 2010; Di Lorenzo et al, 2013). We previously showed that NO in turn regulates MT1-MMP activity during EC migration (Genis et al, 2007), and our new findings thus define an additional layer of feedback NO regulation during this process.

MT1-MMP actions can rely on its catalytic or signaling capabilities (Koziol et al, 2012b). Although we cannot exclude a

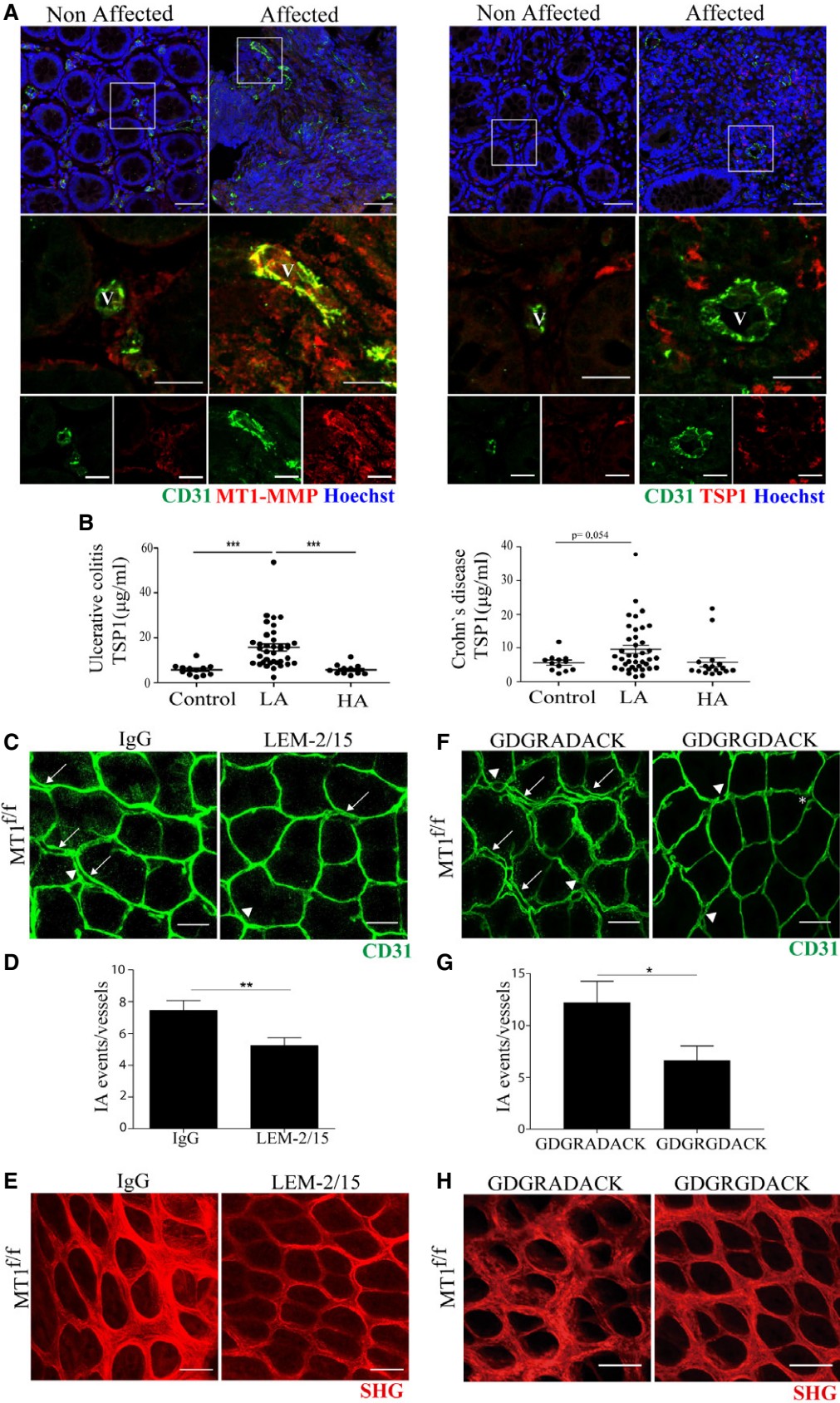

**Figure 8.**

◄

**Figure 8.** **The MT1-MMP/TSP1/αvβ3 integrin pathway is a potential biomarker and therapeutic target in inflammatory bowel disease.**

A   Representative maximum-intensity projection images of staining for CD31 (green), MT1-MMP or TSP1 (red), and Hoechst (blue, nuclei) in colon sections obtained from an IBD patient. Boxed areas are shown in magnified views below with merged images of green and red channels in the middle and single channels to the bottom. Non-affected and affected mucosa areas from the same patient are shown for comparison. Scale bar, 50 μm (upper panel) and 20 μm (magnified views). V, vessel.

B   ELISA analysis of serum TSP1 in healthy individuals and patients affected by ulcerative colitis (left) or Crohn's disease (right). $n = 12–37$ individuals per condition. LA and HA indicate patients with low and high active disease based on clinical score. Data are shown as individual values and mean ± SEM and were tested by one-way ANOVA with Benjamini and Hochberg post-test; ***$P < 0.001$.

C   Representative maximum-intensity projection images of CD31 staining (green) in whole-mount colon mucosal plexus of MT1$^{f/f}$ mice injected intraperitoneally with 3 mg/kg of the anti-MT1-MMP inhibitory antibody LEM-2/15 or its isotype control at day 0 and day 2 during the 3 days of treatment with 1% DSS. Scale bar, 40 μm. Arrows, arrowheads, and asterisks indicate duplications, loops, and pillars, respectively.

D   Quantification of IA events in mice treated as in (C); $n = 8$ mice per condition in two independent experiments. Data are shown as mean ± SEM and were tested by t-test; **$P < 0.01$.

E   Representative second-harmonic generation (SHG) microscopy images of whole-mount colons from MT1$^{f/f}$ mice treated as in (C). Scale bar, 40 μm.

F   Representative maximum-intensity projection images of CD31 staining (green) in whole-mount colon mucosal plexus of MT1$^{f/f}$ mice treated with the TSP1 nonamer GDGRGDACK or its control peptide GDGRADACK by continuous minipump delivery (up to 2.4 mg/mouse/day) for 3 days during 1% DSS treatment. Scale bar, 40 μm. Arrows, arrowheads, and asterisks indicate duplications, loops, and pillars, respectively. Mice/n independent experiments.

G   Quantification of IA events in mice treated as in (F); $n = 5$ and 6 mice per condition in two independent experiments. Data are shown as mean ± SEM and were tested by t-test; *$P < 0.05$.

H   Representative second-harmonic generation (SHG) microscopy images of whole-mount colons from MT1$^{f/f}$ mice treated as in (F). Scale bar, 40 μm.

Data information: Please see Appendix Table S3 for exact P-values.
Source data are available online for this figure.

contribution to IA through mechanisms involving regulation of Rac1 signaling activity (Gonzalo et al, 2010), the lentivirus in vivo rescue experiments indicate the involvement of MT1-MMP catalytic activity. Our earlier work identified TSP1 as the master substrate nucleating the combinatorial proteolytic MT1-MMP program related to EC migration, invasion, and vessel formation in endothelial cells stimulated with TNFα (Koziol et al, 2012a), a crucial cytokine during colitis progression (Danese, 2008). TSP1 is a pleiotropic matricellular protein widely recognized as an inhibitor of angiogenesis, although certain domains can stimulate ECs (Armstrong & Bornstein, 2003). TSP1 is therefore considered a promising target in cancer and inflammatory and vascular diseases, and agonist and antagonist peptide sequences and domains have been trialed in animal models (Taraboletti et al, 2010; Lopez-Ramirez et al, 2017). TSP1 expression has been reported in the context of vascular expansion by IA, for example, in the chorioallantoid membrane and during colitis; trials of the TSP1 mimetic octapeptide ABT-898 in the mouse colitis model showed a partial decrease in microvascular density (Gutierrez et al, 2015). Nevertheless, to our knowledge, the role of TSP1 in IA, particularly in the inflammatory context, has not yet been directly assessed.

Interaction of TSP1 with CD36 via its type I repeat and with CD47 via its C-terminal globular domain inhibits NO signaling through several mechanisms (Isenberg et al, 2008a). Our data reveal that TSP1 processing by MT1-MMP contributes to NO production. Cleavage of TSP1 by MT1-MMP might disturb CD36 binding and unbrake its NO inhibition (Isenberg et al, 2009). However, work from Iruela-Arispe's group on TSP1 processing by Adamts1 during wound healing predicts that the cleaved N-terminal TSP1 fragment will be unstable, whereas the C-terminal fragment (able to bind CD47/αvβ3 integrin) can remain in the inflamed tissue and be released to the serum (Lee et al, 2006). Our results confirm the presence of TSP1 in the serum of both mice and patients affected by colitis. Moreover, we demonstrated that NO production is induced by the C-terminal TSP1 fragment in MT1-MMP-deficient ECs, and contrarily, it is reduced in MT1-MMP-expressing ECs by inhibiting the interaction of TSP1 with αvβ3 integrin or CD47 (Isenberg et al,

2008b), which is known to associate with and regulate β3 integrin (Chung et al, 1997). Our data indicate that MT1-MMP-mediated cleavage of TSP1 releases TSP1 fragments from the matrix, allowing their interaction with the CD47/αvβ3 integrin complex likely active at the dorsal plasma membrane (Lindberg et al, 1996). Our results also identify for the first time that interaction of αvβ3 integrin with the RGD sequence in TSP1 is crucial to the induction of NO production in vitro and inflammatory IA in vivo. αvβ3 integrin has long been recognized as a crucial regulator of vascular development and angiogenesis, with fine-tuned actions depending on the initial status of the vasculature and/or the accompanying molecular microenvironment, similar to the dual actions described for TSP1 (Armstrong & Bornstein, 2003; Robinson & Hodivala-Dilke, 2011). Danese and coworkers previously showed that treatment with the non-RGD-based peptide ATN161, which blocks both αvβ3 and α5β1 integrins, effectively decreased angiogenesis and improved colitis in the IL-$10^{-/-}$ mouse model but not in the DSS-induced model (Danese et al, 2007a). Our study supports integrin specificity, demonstrating that αvβ3 integrin binding to TSP1 drives NO production in vitro and IA in vivo. This is likely related to the induction of selective signals by αvβ3 but not β1 integrins (Leavesley et al, 1993; Yurdagul et al, 2013). Whether TSP1 interactions with other receptors in ECs (such as calcium channels) play additional roles in NO production will require further research (Resovi et al, 2014; Risher et al, 2018). We propose that the proteolytic processing of TSP1 enables pleiotropic actions (Iruela-Arispe, 2008). Upregulated MT1-MMP and TSP1 expression in inflammation, vascular disease, and cancer will favor cleaved TSP1-αvβ3 integrin interaction, NO production, and IA in preference to TSP1-CD36 binding, macrophage recruitment, and inflammation (Lopez-Dee et al, 2011). Since ECs express low levels of MT1-MMP and TSP1 under homeostatic conditions, the upregulation of endothelial MT1-MMP and thus TSP1 processing in specific disease settings may account for the reported dose or cell-type-dependent dual actions of TSP1 on NO and inflammation (Armstrong & Bornstein, 2003).

Although promotion of lymphangiogenesis by VEGF-C seems a promising strategy for colitis alleviation in mouse models

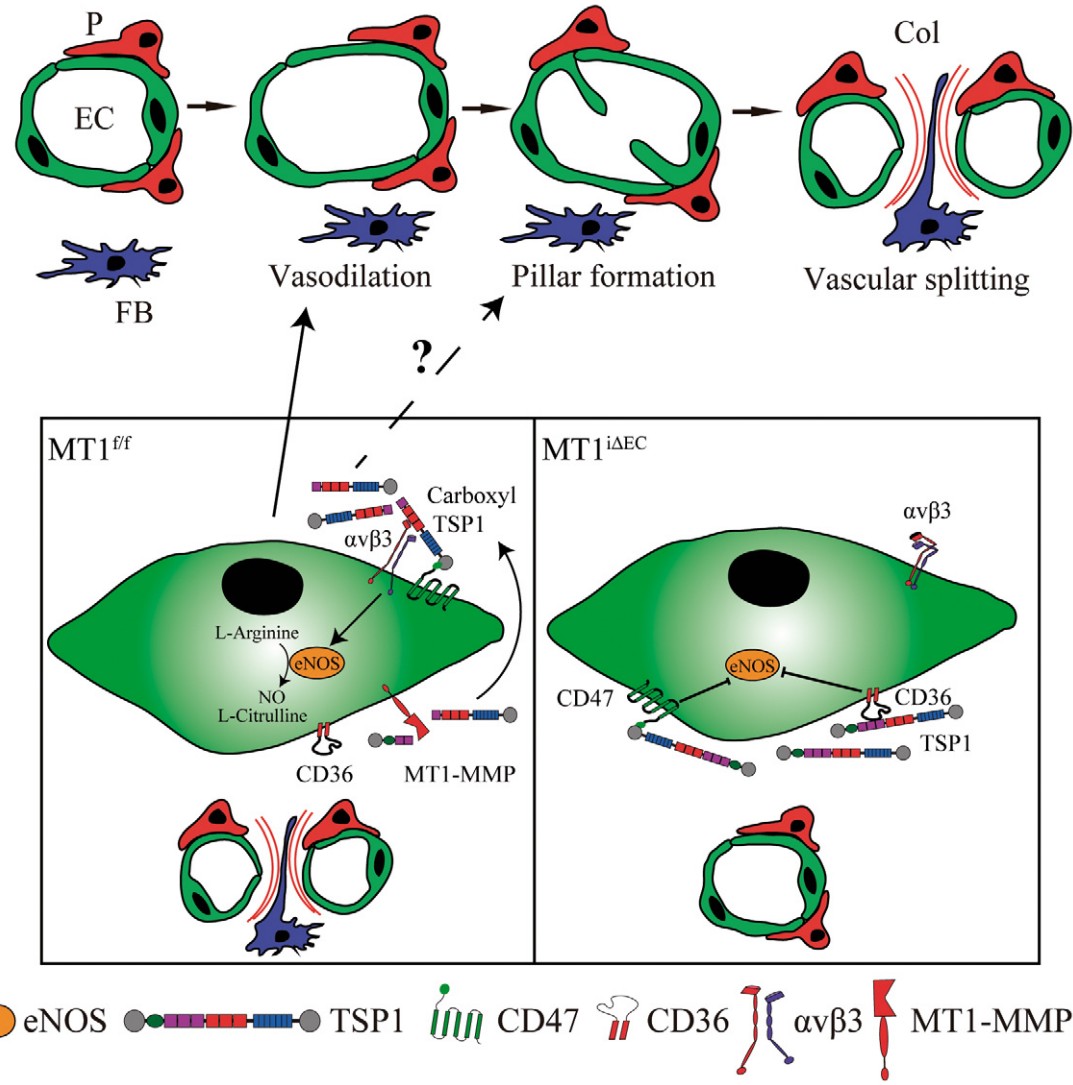

**Figure 9. Working model of the role of MT1-MMP/TSP1/nitric oxide axis in IA during colitis.**

The scheme depicts the role that endothelial cell MT1-MMP (left) exerts in IA during colitis by processing TSP1 and binding of the generated TSP1 C-terminal fragment to CD47/αvβ3 integrin, thus inducing nitric oxide production. This signaling cascade ultimately leads to arteriole vasodilation and endothelial cell intussusceptive remodeling in the capillary plexus of the inflamed intestinal mucosa. When MT1-MMP is absent from endothelial cells (right), this pathway is impaired, IA reduced, and colitis ameliorated. EC, endothelial cell; P, pericyte; FB, fibroblast; Col, collagen.

(D'Alessio *et al*, 2014), there is still no proven benefit for IBD patients of targeting angiogenesis, whether indirectly (inhibiting the immune response) or directly (inhibiting VEGF or other angiogenic molecules as TNFα) (Chidlow *et al*, 2006; Eder *et al*, 2015). In this regard, conflicting effects have been reported for VEGF inhibition in colitis, likely related to the kinetics and efficiency of its targeting (Tolstanova *et al*, 2009; Chernoguz *et al*, 2012). This controversy underlines the need to deepen our understanding of the mechanisms underlying IA in IBD for a more rational design of angiogenesis inhibitors. Our findings identify the MT1-MMP/TSP1/ αvβ3 integrin/NO pathway as a possible therapeutic target for IA and colitis since blocking this axis by inhibiting MT1-MMP activity or competing TSP1 binding to αvβ3 integrin both diminished IA and preserved collagen fiber organization. Moreover, the *in silico*

docking of the TSP1 cleavage sites with MT1-MMP could permit small-molecule screening and buildup to cell-specific strategies involving peptide-directed delivery, nanoparticles, or other modes of tissue-tailored therapeutic intervention. It is remarkable that the elevated serum TSP1 detected in patients with low-activity colitis, but not high-activity disease, correlated with the protection conferred by MT1-MMP endothelial deletion in mild mouse colitis but not severe disease (not shown). This suggests that the MT1-MMP/TSP1/NO pathway may be particularly important in early stages or mild forms of the disease, where the IA vascular response plays a predominant role in colitis progression. Finally, our data from IBD-affected patients showing elevated MT1-MMP expression in the inflamed vasculature and elevated serum TSP1 suggest their potential usefulness as surrogate biomarkers of disease activity.

## Materials and Methods

### Animals

C57BL/6 wild-type mice were purchased from Charles River, and MT1-MMP (*Mmp14*)-lacZ reporter mice were as previously reported (Yana *et al*, 2007). *Mmp14*$^{loxP/loxP}$ mice (Gutierrez-Fernandez *et al*, 2015) were crossed with *Cdh5*-Cre$^{ERT2}$ mice (kindly provided by Prof. Adams' laboratory; Wang *et al*, 2010) to generate *Mmp14*$^{loxP/loxP}$;*Cdh5*-Cre$^{ERT2/+}$ mice (MT1$^{iΔEC}$ mice). Mice at 8–20 weeks old were injected intraperitoneally with 1 mg per mouse of 4-hydroxytamoxifen (4-OHT) on five consecutive days to induce endothelial cell-specific deletion of MT1-MMP (Monvoisin *et al*, 2006). Mice were housed and all animal experiments performed under specific pathogen-free conditions at the Animal facility of Centro Nacional de Investigaciones Cardiovasculares Carlos III (CNIC), and in strict accordance with the institutional guidelines. Mice were kept under a 12-h light/dark cycle (lights on from 07:00 to 19:00 h) and fed *ad libitum* with standard chow diet (2108 Teklad Global, Harlan Interfauna Iberica S.L.). Mice were genotyped by tail DNA PCR using the following primers: for the MT1-flox recombination, 5′-CCCTGGG TCAACTACAGCA-3′ and 5′-TTTGTGGGTGACCCTGACTTG-3′; for LacZ, 5′-ATCGTGCGGTGGTTGAA-3′ and 5′-TGCTGACGGTTAACGC CTCG-3′; and for Cre, 5′-AGGTGTAGAGAAGGCACTTAGC-3′ and 5′-CTAATCGCCATCTTCCAGCAGG-3′.

### Mouse experimental colitis

Experimental colitis was induced in 8- to 20-week-old mice by supplying 1% dextran sodium sulfate (DSS) (MP Biomedicals, 0216011010) dissolved in drinking water *ad libitum* for 3 or 7 days. Daily monitoring included measuring body weight and calculating disease activity index (DAI). The baseline colitis on day 0 was scored as 0. Subsequent weight loss of 1–5% was scored as 1, 5–10% as 2, 10–20% as 3, and more than 20% as 4. For stool consistency, normal, well-formed pellets were scored as 0, semi-formed pellets not adhering to the anus were scored as 2, and liquid stools adhering to the anus were scored as 4. Similarly, no bleeding was assigned a score of 0, positive blood detection in the stool was scored as 2 (detected by hemoFEC, Cobas, 10243744), and gross rectal bleeding was scored as 4. At the end of the DSS exposure period, the whole colon was removed and cleaned with PBS. The most distal portion was fixed in 4% PFA overnight and then used directly for whole-mount staining, processed, and embedded in paraffin or frozen in OCT for histological analysis. The remaining part of the colon was homogenized for protein and RNA extraction. The anti-MT1-MMP antibody LEM-2/15 and the E123CaG-1 TSP1 fragment were administered to mice by intraperitoneal injection at 3 mg/kg and 1 μg, respectively, on day 0 and day 2 during the 3 days of 1% DSS treatment. No statistical methods were used to pre-estimate the animal sample size, and mice were randomly allocated to experimental groups. Investigators were not blinded during the analysis of mouse samples.

### Lentiviral infection

The full-length MT1-MMP sequence (FL) or mutated versions to disable catalytic activity (E240A, EA) or block signaling activity (Y573F, YF) were cloned into the SFFV-IRES-GFP lentiviral backbone. Lentiviruses expressing Mock, MT1-MMP FL, EA, or YF were prepared and titered as previously described (Martin-Alonso *et al*, 2015). Lentivirus solution (100 μl) was injected intravenously into the jugular vein of adult MT1$^{iΔEC}$ mice at $1 × 10^7$ pfu/ml. Mice were administered 1% DSS for 3 days and then sacrificed, and colons were processed for whole-mount staining.

### Subcutaneous pump implantation

MT1-MMP$^{f/f}$ control mice were anesthetized with isoflurane and implanted with subcutaneous micro-osmotic pumps (Alzet, model 10003D) charged with 10 mg/100 μl of the peptides GDGRGDACK or GDGRADACK. Pumps released ~1 μl/h (100 μg/h), equivalent to 2.4 mg/mouse/day. Mice were administered 1% DSS for 3 days and then sacrificed, and colons were processed for whole-mount staining.

### Endothelial cell culture

Human umbilical vein endothelial cells (HUVEC) were purchased from Lonza (BioWhittaker, CC-2519) and cultured on 0.5% gelatin-coated plates (unless otherwise indicated) in medium 199 supplemented with 20% FBS, 100 U/ml penicillin (Lonza), 100 μg/ml streptomycin (Lonza), 2 mM L-glutamine (Lonza), 10 mM HEPES (Lonza), and ECGS/H (Promocell, C-30120). Cells were used up to passage 5. Mouse aortic endothelial cells (MAEC) were obtained from aortas of 4-week-old mice. Briefly, after fat removal under a microscope, aortas were incubated for 5 min at 37°C in collagenase solution (collagenase type I, 3.33 mg/ml, Worthington, LS004194), thus allowing removal of the adventitia with forceps. The aortas were then cut into small pieces (1–2 mm), and a cell suspension was obtained by incubation for 45 min at 37°C in 6 mg/ml type I collagenase and 2.5 mg/ml elastase (Worthington, LS002290) diluted in DMEM. Once EC colonies were visible, they were subjected to two rounds of positive selection with anti-ICAM2 and magnetic beads. MAEC were cultured on 0.2% gelatin-coated plates in DMEM/F12 medium supplemented with 20% FBS, 100 U/ml penicillin (Lonza), 100 μg/ml streptomycin (Lonza), 2 mM L-glutamine (Lonza), 10 mM HEPES (Lonza), and ECGS/H (Promocell, C-30120).

### MT1-MMP siRNA silencing

HUVEC were transfected twice with 10 nM MT1-MMP-specific siRNAs [Ambion, Thermo Fisher Scientific, MT1-ARNi#1, 8877 (4390824); MT1-ARNi#2, 8879 (4390824)] or control scramble siRNA [Ambion, Thermo Fisher Scientific, Control No. 1 (4390843)] using RNAiMAX Lipofectamine (Thermo Fisher Scientific, 13778030). Downregulation was confirmed by qPCR and Western blot.

### Vascular perfusion and permeability assays

For vascular perfusion analysis, mice received intravenous injections of 100 μl of biotinylated or FITC-GSL I Isolectin B4 (Vector Labs, B1205 and FL-1201, respectively) 20 min before sacrifice. Mice were perfused under constant pressure with cold PBS, and colon samples were extracted and processed for whole-mount staining. Vascular perfusion was quantified as the percentage of the CD31-positive vascular volume containing an IB4 signal.

For vascular permeability analysis, mice received intravenous injections of 100 μl of 0.5 mg/ml (71.5 μM) Dextran-TRITC 70 KD (Invitrogen, D1819) 20 min before sacrifice. Colon samples were extracted and frozen in OCT, and 30-μm-thick sections were immunostained and visualized.

## Quantitative real-time PCR

RNA was extracted with the RNeasy Plus Micro Kit (Qiagen, 74034). cDNA was synthesized using the High-Capacity cDNA Reverse Transcription Kit (Applied Biosystems, 4368814). Quantitative PCR was performed using Power SYBR Green PCR Master Mix (Applied Biosystems, 4367659) following the manufacturer's protocol. The primers used in qPCR are listed in Appendix Table S2. GAPDH, 36b4, and TBP were used as internal expression controls. The results (CNRQ; calibrated normalized relative quantity) were analyzed with qbase Plus software (Biogazelle, Gent, Belgium).

## Western blot

Cells were lysed in RIPA buffer supplemented with protease and phosphatase inhibitors. Protein samples were separated by 8% SDS–PAGE, transferred to nitrocellulose membranes, blocked with 5% BSA, and incubated sequentially with primary and horseradish peroxidase (HRP)-conjugated secondary antibodies, with washes. The primary antibodies used were anti-MT1-MMP (LEM-2/15) (Galvez et al, 2001), anti-eNOS (BD Bioscience, 610297), anti-GFP (Abcam, ab13970), anti-GAPDH (Sigma-Aldrich, G9545), and anti-β-actin (Sigma-Aldrich, A5441). All primary antibodies were used at a dilution of 1:1,000. The secondary antibodies used were HRP goat anti-mouse and HRP goat anti-rabbit (Jackson), and bound secondary antibodies were visualized with Luminata Classico Western HRP Substrate (Millipore, WBLUC0500) in ImageQuant LAS 4000 (GE Healthcare Life Sciences, Massachusetts, USA). Western blots were quantified using ImageJ software (https://imagej.nih.gov/ij/).

## In vitro digestion assay

Human TSP1 purified from plasma (a gift from Dr. G. Taraboletti, Istituto di Ricerche Farmacologiche Mario Negri IRCCS, Bergamo, Italy) was incubated with increasing amounts of the catalytic domain of human recombinant MMP14 (hrMMP14, ref. 475935; Calbiochem, San Diego, USA) in digestion buffer (50 mM Tris–HCl, 10 mM CaCl2, 80 mM NaCl [pH 7.4]) for 2 h at 37°C. Samples were separated by 10% SDS–PAGE and transferred to nitrocellulose membranes (Bio-Rad, 45 μm). Full-length TSP1 and fragments were detected with an anti-TSP1 antibody recognizing an epitope nearby the N-terminus (Lee et al, 2006). An anti-rabbit HRP secondary antibody (Jackson) was used and the bound proteins developed with Luminata Classico Western HRP Substrate (Millipore, WBLUC0500) in ImageQuant LAS 4000 (GE Healthcare Life Sciences, Massachusetts, USA).

## Whole-mount staining

For cremaster muscle, anesthetized mice were intravenously injected with 100 μl of 100 μM acetylcholine 15 min before sacrifice and with 100 μl of 100 μM sodium 2-(N,N-diethylamino)-

diazenolate-2-oxide (DEANO) 5 min before sacrifice; saline was used as a control for both injections. For VEGF administration to the distal colon, 20 ng of VEGF (PeproTech, 450-32) in a final volume of 50 μl was delivered into anesthetized mice via a rectal cannula, 10 min before sacrifice. Cremaster muscle or distal colon was then removed, cleaned, flat-mounted, and fixed with 4% PFA overnight at 4°C. Mouse and human whole-mount samples were blocked and permeabilized in blocking solution (PBS 5% BSA, 5% normal goat serum, and 0.3% Triton X-100). Primary antibodies were diluted 1:100 in blocking solution, and samples were incubated on a shaker at 4°C overnight. Vessels were stained with anti-CD31 (Millipore, MAB1398Z), anti-ICAM2 (BD Pharmingen, 553325), and anti-ERG 647 (Abcam, ab196149). The following day, the samples were washed 3 × 1 h in PBS and 0.1% Triton X-100 and incubated overnight at 4°C with the appropriate secondary fluorescent antibodies (goat anti-Armenian hamster IgG, anti-mouse IgG, anti-rabbit IgG, or anti-rat IgG, all from Jackson ImmunoResearch) and Hoechst 33342 (Invitrogen, H1399). The next day, samples were washed 3 × 1 h in PBS and 0.1% Triton X-100 and mounted in glycerol. Fluorescence microscopy images were acquired with a Zeiss LSM 700 confocal microscope at 21–23°C using Plan-Apochromat, 25 × 0.8 or 63× glycerol DIC M27, and z-stacks were captured every 1.5 or 1 μm, respectively. Confocal images were imported into Imaris (version 7.7.2; Bitplane AG, Switzerland) to obtain a precise 3D reconstruction generated by the program's iso-surface rendering function of this software. In 3D images of the mucosal vasculature, we evaluated vascular volume, intussusception events (holes, loops, and duplications), and vascular perfusion, and in 3D images of the cremaster muscle, capillary diameters were quantitated. Intercapillary angles in the mucosa vascular plexus were measured by ImageJ software. Capillary diameters in CD31-stained whole-mount colons after VEGF rectal administration were quantified by an ImageJ ad-hoc plugin (http://adm.irbbarcelona.org/image-j-fiji#TOC-Blood-ve ssel-segmentation-and-network-analysis).

## Second-harmonic generation multi-photon microscopy

Autofluorescence of fibrillar collagen (425- to 465-nm filter) was visualized in flat-mounted fixed colon from MT1^f/f and MT1^iΔEC mice with a Zeiss LSM 780 microscope coupled to a Spectra-Physics Mai Tai DS Laser, using an 800 nm excitation wavelength, which resulted in the SHG signal detectable at 400 nm (emission fluorescence range 388–408 nm).

## Endothelial cell immunofluorescence staining

Confluent ECs on glass coated with 1% gelatin were fixed with 4% PFA during 8 min at RT. The cells were stained with anti-MT1-MMP (LEM-2/15) and anti-TSP1 (Thermo Fisher, MA5-13398) primary antibodies diluted 1:100. Cells were incubated with appropriate secondary fluorescent antibodies (anti-mouse IgG, anti-rabbit IgG from Jackson ImmunoResearch), phalloidin FITC (Life Technologies, F432), and Hoechst 33342 (Invitrogen, H1399) for 30 min at 37°C. Fluorescence microscopy images were acquired with a Zeiss LSM 700 confocal microscope at 21–23°C using Plan-Apochromat, 40 × 0.8 oil DIC M27, and z-stacks were captured every 1 μm. Images were analyzed using ImageJ software (https://imagej.nih.gov/ij/).

## Cell sorting

Freshly isolated mouse lung endothelial cells (MLECs) were obtained from lungs as described (Oblander et al, 2005). Red blood cells were removed by incubating the lung cell suspension for 5 min in ACK buffer (150 mM $NH_4Cl$, 0.1 mM $Na_2EDTA$, 10 mM $KHCO_3$, pH = 7.4). The cells were then resuspended in sorting buffer (PBS + 1% FBS + 10 mM HEPES) and incubated for 10 min at 4°C with an Fc-block anti-CD16/CD32 mAb (Pharmingen) to block non-antigen-specific binding of immunoglobulins to the Fcγ receptors. This was followed by staining with Alexa Fluor 488-conjugated anti-mouse CD31 (Mec13.3) and V405-conjugated anti-mouse CD45.1 (all from BD Bioscience Pharmingen). The endothelial cell population defined as $CD31^+/CD45^-$ was sorted in a Synergy 4L Cell Sorter (Sony Biotechnology Inc.), and MT1-MMP deletion efficiency was analyzed by qPCR.

## Enzyme-linked immunosorbent assay

C57BL/6 mice were administered 1% DSS for 7 days; blood was collected by cardiac puncture, and serum was separated and stored at −80°C. Soluble TSP1 was measured in serum by enzyme-linked immunosorbent assay (Mouse TSP1 ELISA kit; Cusabio, CSB-E08765m). Soluble TSP1 in serum from human control individuals and IBD patients was measured by ELISA (Human TSP1 DuoSet ELISA; R&D Systems, DY3074). Mean optical density (OD) at 450 nm was calculated from triplicate readings per sample, and OD values at 570 nm were subtracted to correct for optical imperfections in the plate reading. A 4-parameter logistic (4-PL) curve fit was generated using Benchmark Plus Microplate Spectrophotometer (BIO-RAD) coupled to Microplate Manager 5.2.1.

## Colon immunofluorescence staining

Paraffin-embedded colonic mouse experimental colitis samples were cut into 5-μm sections, deparaffinized, hydrated through an alcohol series of decreasing concentration, and treated with sodium citrate buffer (10 mM sodium citrate, 0.05% Tween-20, pH 6.0) for 20 min at 95°C and then cooled for 2 h at room temperature for antigen retrieval. Sections were permeabilized with PBS containing 0.3% Triton X-100 for 10 min at room temperature and blocked with Image-iT™ FX signal enhancer (Invitrogen, 36933) for 30 min at room temperature and with blocking buffer (PBS with 5% BSA and 5% goat serum) for 1 h at room temperature. Sections were incubated with the following primary antibodies overnight at 4°C in blocking buffer: anti-CD31 (DIANOVA, DIA310) anti-MT1-MMP (LEM-2/15) and anti-TSP1 (provided by Dr M. L. Iruela-Arispe, University of California, Los Angeles, CA, USA, 10 μg/ml). Subsequently, samples were washed with PBS and 0.1% Triton X-100 and incubated for 2 h at room temperature with the corresponding conjugated secondary antibodies and Hoechst 33342. Samples were mounted in Fluoromount-G imaging medium (4958-02, Affymetrix eBioscience).

Samples included in OCT were permeabilized and blocked in PBS containing 0.3% Triton X-100, 5% BSA, 5% goat serum, and anti-CD16/CD32 (clone 24g2, 1:100) for 1 h at room temperature. Samples were then incubated overnight at 4°C with anti-CD11b Alexa 647 (clone M1/70, eBiosciences, 51-0112-82), anti-Erg, anti-β-galactosidase, and anti-ICAM2. Samples were then incubated with the corresponding secondary antibodies and Hoechst 33342 for 2 h at room temperature. Finally, samples were mounted in Fluoromount-G, and images were acquired with a Zeiss LSM 700 confocal microscope fitted with a 25× objective and processed and quantified using ImageJ software.

## Nitric oxide production

Confluent ECs were seeded in 96-well glass plates coated with 1% gelatin and cultured in EGM2 (Promocell, C-22111). The cells were incubated with 2.5 μM DAF-FM diacetate (Life Technologies, D23842) for 30 min at 37°C and washed with PBS. The DAF-FM fluorescence signal was visualized with a Zeiss LSM 700 confocal microscope (Plan-Apochromat 40 × 1.3 Oil DIC M27), and mean fluorescence intensity (MFI) at single-cell level was analyzed with ImageJ software. HUVEC were treated for 24 h with antibodies to MT1-MMP (LEM-2/15), integrin αv [anti-αv integrins ABA 6D1 (Yanez-Mo et al, 1998)], or CD47 (Abcam, ab3283), with a range of concentrations of the peptides RGDS, RADS, cilengitide, TSP1 nonamer GDGRGDACK, or the control GDGRADACK, or with human full-length TSP1 or the human recombinant TSP1 fragment E123GaC-1.

## Intravital microscopy in the cremaster muscle

For the analysis of arteriolar vasodilation, mice were anesthetized, and the cremaster muscle was isolated and exposed on the microscope plate as described (Rius & Sanz, 2015). Mice were injected with 100 μl of 100 μM acetylcholine, and arterioles were recorded for at least 15 min using a Leica DM6000-FS intravital microscope fitted with an Apo 40× NA 1.0 water-immersion objective and linked to a DFC350-FX camera. LASAF software was used for image acquisition and processing.

## In silico modeling of MT1-MMP dimer

The MT1-MMP monomer was modeled by submitting the Fasta sequence of mature human MT1-MMP protein (UniProt ID: P50281, residues 112–582) to a local implementation of I-Tasser software suite v4.4 (Yang et al, 2015) for threading modeling with homology. The selected model was that with minimal energy and correct folding (best structural alignment to templates; PDB ID: 1BQQ for catalytic domain and 3C7X for hemopexin-like domain). In this model, the transmembrane domain (TM) is buried in the protein core. To fix this, this region (transmembrane plus C-term, TMIC) was extracted from the model, angles were fixed according to the predicted secondary structure, and a refinement cycle was performed using the relax tool (Nivon et al, 2013; Conway et al, 2014) in Rosetta suite v3.5 release 2015.38.58158 (www.rosettacommons.org). The best model for TMIC (lower energy and best stability) was selected and aligned with the previous MT1-MMP model, and the new model composed of MT1-MMP without TMIC domains and the new TMIC model was used as the template; the gap between them was closed using the loopmodel tool (Mandell & Kortemme, 2009; Stein & Kortemme, 2013) in Rosetta suite. The most stable model was selected as the final model for the MT1-MMP monomer. To

## The paper explained

### Problem

Human inflammatory bowel disease (IBD) is a chronic inflammatory disease of the intestine comprising ulcerative colitis and Crohn's disease and characterized by phases of remission and relapse. IBD is a multifactorial disease featuring a primary defect in intestinal epithelial barrier integrity and an exacerbated immune response to the microbiota. The use of immunomodulators, new biologics, and small molecules has improved IBD treatment in the recent years, but non-responder patients and the unpredicted nature of relapses make necessary to expand our knowledge about other processes influencing IBD pathophysiology. Colitis progression seems to involve angiogenesis, the formation of new vessels from pre-existing ones, which is regarded as a potential therapeutic target but with limited success in animal models. A better understanding of the molecular pathways behind colitis-associated angiogenesis will help designing more efficient and tailor-based therapies for this complex disease.

### Results

In this study, we implemented confocal microscopy and 3D image reconstruction tools to investigate intussusceptive (or splitting) angiogenesis (IA), a poorly characterized mode of angiogenesis, in a mouse model of IBD. After induction of mild colitis, we detected an increased number of vascular events, all hallmarks of IA. We next explored the role of the matrix metalloproteinase MT1-MMP in IA and we demonstrated that the selective absence of MT1-MMP from endothelial cells decreased IA events and led to improved vascular perfusion, milder clinical score, and better-preserved intestine morphology during mouse colitis.

Vasodilation is an obliged prerequisite to IA, and we demonstrated that endothelial MT1-MMP absence decreased capillary vasodilation induced by VEGF in the intestine and by acetylcholine in the muscle. We showed *in vivo* and *in vitro* that MT1-MMP catalytic activity was required for nitric oxide production and for the induction of IA. *In vitro* digestion assays, *in silico* protein modeling, and endothelial cell culture identified the cleavage of the matricellular protein thrombospondin-1 (TSP1) by MT1-MMP as responsible for nitric oxide production. Finally, we found higher levels of TSP1 in serum from low active IBD patients, and we reduced IA during mouse colitis progression via anti-MT1-MMP antibody injection or delivery of a blocking TSP1 peptide.

### Impact

We propose that our findings have both diagnostic and therapeutic clinical impacts for patients with IBD. Indeed, our data from IBD-affected patients suggest that MT1-MMP and TSP1 could serve as surrogate biomarkers of disease activity. Moreover, our findings identify the MT1-MMP/TSP1/αvβ3 integrin/nitric oxide pathway as a novel actor in IA whose targeting ameliorates colitis progression. We have demonstrated this therapeutic application by blocking this axis with anti-MT1-MMP inhibitory antibodies, and through selective competition of TSP1 binding to αvβ3 integrin by continuous peptide delivery. Since we found elevated serum TSP1 in patients with low active colitis and MT1-MMP endothelial deletion protected mainly from mild colitis, these therapeutic options may be of special value in early stages or mild forms of IBD. Finally, the *in silico* docking of the TSP1 cleavage sites with MT1-MMP could lead to small-molecule screening and to new cell-specific strategies involving peptide-directed delivery, nanoparticles, or other modes of tissue-tailored therapeutic intervention.

model dimeric MT1-MMP, two of these monomers were positioned according to the published MT1-MMP dimer interface using PyMOL v1.8 (www.pymol.org), and the new dimer model was used as the initial template. A new PDB file was generated for the model, in which the membrane protein structure is transformed into PDB coordinates (with the *z*-axis normal to the membrane) using the PPM server (http://opm.phar.umich.edu/server.php). A full spanfile was generated from the PDB structure using the spanfile_from_pdb application in the membrane framework in Rosetta suite v3.5 release 2015.38.58158. An initial relax cycle was performed using the relax application in Rosetta suite to minimize energy and clashes using the initial template and the full spanfile (Conway *et al*, 2014). The lowest scoring refined model (lowest energy) was selected as the input model, and a new dimer model was generated using the mp_dock application in the Rosetta suite membrane framework (Alford *et al*, 2015). All models with TM embedded in the membrane clustered together. The best model, with lowest energy, was selected as the final dimer. This homodimer meets the previously described binding interface constraints, with docking via the hemopexin-like domains, and has quasi-C2 symmetry. We next positioned the MT1-MMP dimer model together with the reported crystal structure of the TSP1 type 1 repeat (PDB ID: 1LSL) using PyMOL and evaluated the global topography and accessibility of the complex.

### Statistics

Protein expression on Western blots was quantified densitometrically using ImageJ software (NIH, Bethesda, USA) and normalized to the corresponding loading control. Data were tested for normal distribution by D'Agostino–Pearson test and compared with non-parametric or parametric statistical tests as appropriate (see figure legends for details). Graphs were prepared and statistical analysis performed using GraphPad Prism 5.0 (GraphPad Software, La Jolla, USA). Data are presented as mean $\pm$ standard error of the mean (SEM). Differences were considered significant at $*P < 0.05$, $**P < 0.01$, $***P < 0.001$, and $****P < 0.0001$.

### Study approval

Animal procedures were approved by the Committee on the Ethics of Animal Experiments of the CNIC (Permit Number: CNIC-01/13) and by the corresponding legal authority of the local government "Comunidad Autónoma" of Madrid (Permit Number: PROEX 34/13). Animal studies were conformed to directive 2010/63EU and recommendation 2007/526/EC regarding the protection of animals used for experimental and other scientific purposes, enforced in Spanish law under RD1201/2005. Human samples were obtained with informed consent after ethical approval by the Hospital de La Princesa Ethics Committee. Informed consent was obtained from all subjects, and the experiments conformed to the principles set out in the WMA Declaration of Helsinki and the Department of Health and Human Services Belmont Report.

**Expanded View** for this article is available online.

### Acknowledgements

We thank Ralf Adams (Max Planck Institute for Molecular Biomedicine, Germany) and Carlos López-Otín for providing the *Cdh5*(BAC)Cre[ERT2] and *Mmp14*[floxed/floxed] mouse lines, respectively; María José Calzada for providing anti-CD47 antibody and human TSP1; Giulia Taraboletti and Deane F. Mosher for providing full-length TSP1 and the E123CaG-1 fragment, respectively;

Ángela Pollán, Ángel Colmenar, and Alberto Jiménez-Montiel for technical support; and Simon Bartlett for English editing. This study was supported by grants from the Spanish Ministerio de Ciencia, Innovación y Universidades (SAF2014-52050-R and SAF2017-83229-R to A.G.A). S.E. was recipient of a FPI-Severo Ochoa fellowship. The CNIC is supported by the Instituto de Salud Carlos III, the Spanish Ministry of Ciencia, Innovación y Universidades, and the Pro-CNIC Foundation and is a Severo Ochoa Center of Excellence (SEV-2015-0505).

## Author contributions

SE performed and analyzed HUVEC experiments and most colitis experiments; CC performed histology, immunofluorescence, *in vitro* digestion assay, some colitis experiments, and image quantification; AK and PG performed and analyzed initial colitis experiments; CR and VA performed and analyzed intravital microscopy experiments; FM performed *in silico* protein modeling; AU provided DSS-model expertise; MS provided the MT1[lacz/+] mouse line; PML, MC, and JPG provided human samples from IBD patients; and AGA designed and supervised the research and wrote the manuscript.

## Conflict of interest

M Chaparro has served as a speaker for or has received research or education funding from MSD, Abbvie, Hospira, Pfizer, Takeda, Janssen, Ferring, Shire Pharmaceuticals, Dr Falk Pharma, and Tillotts Pharma. JP Gisbert has served as a speaker, a consultant, and advisory member for or has received research funding from MSD, Abbvie, Hospira, Pfizer, Kern Pharma, Biogen, Takeda, Janssen, Roche, Sandoz, Celgene, Ferring, Faes Farma, Shire Pharmaceuticals, Dr. Falk Pharma, Tillotts Pharma, Chiesi, Casen Fleet, Gebro Pharma, Otsuka Pharmaceutical, and Vifor Pharma. The rest of the authors declare no conflict of interest.

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
