## [Review Process File · EMBO Molecular Medicine]

Endothelial MT1-MMP targeting limits intussusceptive angiogenesis and colitis via TSP1-nitric oxide axis

Sergio Esteban, Cristina Clemente, Agnieszka Kozioł, Pilar Gonzalo, Cristina Rius, Fernando Martínez, Pablo M. Linares, María Chaparro, Ana Urzainqui, Vicente Andrés, Motoharu Seiki, Javier P. Gisbert, and Alicia G. Arroyo

Review timeline:

Submission date:	10 May 2019
Editorial Decision:	3 June 2019
Revision received:	30 September 2019
Editorial Decision:	25 October 2019
Revision received:	5 November 2019
Accepted:	8 November 2019

Editor: Lise Roth

Transaction Report:

1st Editorial Decision

3 June 2019

Thank you for the submission of your manuscript to EMBO Molecular Medicine. We have now received feedback from the three reviewers who agreed to evaluate your manuscript. As you will see from the reports below, the referees acknowledge the interest of the study. However, they also raise substantial concerns on your work, which should be convincingly addressed in a major revision of the present manuscript. In particular, TSP1 cleavage should be clearly demonstrated, controls and explanations should be provided, overstatements should be tuned down, and the manuscript should be edited for language and grammar.

Addressing the reviewers' concerns in full will be necessary for further considering the manuscript in our journal, and acceptance of the manuscript will entail a second round of review. EMBO Molecular Medicine encourages a single round of revision only and therefore, acceptance or rejection of the manuscript will depend on the completeness of your responses included in the next, final version of the manuscript. For this reason, and to save you from any frustrations in the end, I would strongly advise against returning an incomplete revision.

Please note that it is EMBO Molecular Medicine policy to allow only a single round of revision and that, as acceptance or rejection of the manuscript will depend on another round of review, your responses should be as complete as possible.

I look forward to receiving your revised manuscript.

***** Reviewer's comments *****

Referee #1 (Remarks for Author):

The submitted MS describes the mechanism of intussusceptive angiogenesis (IA) in inflammatory bowel disease. The authors claim that mice lacking the protease MT1-MMP in endothelial cells (MT1 Δ EC) have limited IA in the capillary plexus of the colon mucosa after 1% DSS-induced colitis. This was correlated with improved tissue perfusion, preserved intestinal morphology, and milder disease activity index. Combined in vivo intravital microscopy and lentiviral rescue experiments with in vitro cell culture demonstrated that MT1-MMP activity in endothelial cells is required for vasodilation and IA, as well as for nitric oxide production via binding of the MT1-MMP substrate thrombospondin-1 (TSP1) to CD47/ α v β 3 integrin. Over all this is a well presented paper that this reviewer had little to add. Experiments are well controlled, not over interpreted and appropriate statistics applied. The mechanism is established. One thing that I missed in Fig 7 was what does 'DAF-FM' stand for in the image analysis?

Referee #2 (Comments on Novelty/Model System for Author):

Major gaps should be filled before the paper is accepted.
E.g. TSP1 cleavage is not demonstrated, while the entire manuscript rests on this assumption. The role of intussusceptive angiogenesis remains unclear 0 could be direct effect of TSP1 on macrophages.

Referee #2 (Remarks for Author):

The study by Esteban et al investigates the role of MT1-MMP in ulcerative colitis/ Crohn's disease. The authors link increase MT1 production by intestinal endothelial plexus with increased NO production due to TSP1 cleavage, which hampers TSP1 inhibitor action on NO production and therefore intussusceptive angiogenesis (IA). The study is potentially quite interesting and the concept is novel. However, there are gaps that should be filled prior to publication.

In general, manuscript is in need of extensive editing by native English speaker/writer, to eliminate inappropriate word usage. Specific comments are listed below.

1. The description for figure 2 states that the perfusion is inhibited in MT1 delta iEC mice. However, partial perfusion is clearly preserved and this statement should be tempered.
2. Usually vascular abnormalities include vascular leakage (also regulated by TSP) Perfusion using FITC-dextran would answer this question.
3. Fig. 2D shows extensive collagen deposition in F/F mice. However, MT-1 digests multiple collagen types. An explanation should be provided.
4. The differences in IA are clear after 3 days, however there is no clear differences in intestinal morphology and macrophage recruitment. Is it possible that IA is not the driving force behind DCC-induced colitis but merely a correlate? Could IA in MT1 delta iEC mice be restored using exogenous angiogenic factor, e.g. VEGF? A propos, it can be released from the ECM by MT1-MMP. Also, TSP1 can have a direct effect on macrophage recruitment and inflammation, via CD36. These possibilities should be discussed in some depth.
5. AcCh is a bit drastic for induction of vasodilation. VEGF would be a more physiological agent. Vasodilation in cremaster muscle is not quite reflective of IA in the colon.
6. The effects of MT1 antibodies and TSP1 receptors on NO production are quite clear. Would MT1 antibody and TSP1 derived ligands alter IA as well?
7. Fig. 6A-C - the lack of TP1 in the vasculature is an overstatement: TSP1 is clearly detectable and the images do not convey major differences. Co-localization with MT1 would be helpful in f/f mice
8. To demonstrate TSP1 cleavage, it would be helpful to incubate intact TSP1 with MT1 positive and negative EC and run a western blot with antibodies to C and N-termini. Otherwise this remains a mere speculation.
9. Fig. 8, MT1 MMP staining looks unconvincing. Improved quality images are required
10. Supplemental Fig. 8; CD36 binding peptide sequence is GVITRIR, not GNITRIR

Referee #3 (Comments on Novelty/Model System for Author):

The manuscript deals with a clinically important issue.
The involvement of the MT1-MMP-TSP1-NO pathway is novel.

The techniques are state of the art.

The authors use a combination of existing animal models, deletion mutant, lentiviral constitution of MT1-MMP and novel point mutants in MT1-MMP to underpin their findings.

The molecular mechanisms of anti-TNF α antibodies in inflammatory bowel disease are still a matter of debate. The present manuscript provides detailed information how TNF α -mediated induction of MT1-MMP via thrombospondin and NO generation affects vasodilation of capillaries accompanied by intussusceptive angiogenesis.

The manuscript has merit for both better understanding of altered vascularization in colitis and is a welcome contribution increasing our insight in the regulation of intussusceptive angiogenesis in general. Although this pathway may not be the only pathway that is involved in this disease, the authors make plausible that the proposed MT1-MMP-TSP1-integrin $\alpha\beta$ 3-NO pathway is a prerequisite for intussusceptive angiogenesis, and contributes to colitis. The techniques used and new mutants generated are state of the art. The data on the accumulation of circulating TSP1 in early colitis patients supports these data, while the lack of such effect in severe patients also points to a complex picture, in particular when one considers colitis treatment via the proposed pathway. Notwithstanding, the manuscript combines new mechanistic insight with a perspective for improved patient care.

I had only a few very minor remarks and would like to recommend the manuscript or publication in EMBO Molecular Medicine.

Referee #3 (Remarks for Author):

The molecular mechanisms of anti-TNF α antibodies in inflammatory bowel disease are still a matter of debate. The present manuscript provides detailed information how TNF α -mediated induction of MT1-MMP via thrombospondin and NO generation affects vasodilation of capillaries accompanied by intussusceptive angiogenesis. The manuscript has merit for both better understanding of altered vascularization in colitis and is a welcome contribution increasing our insight in the regulation of intussusceptive angiogenesis in general. Although this pathway may not be the only pathway that is involved in this disease, the authors make plausible that the proposed MT1-MMP-TSP1-integrin $\alpha\beta$ 3-NO pathway is a prerequisite for intussusceptive angiogenesis, which contributes to colitis. The techniques used and new mutants generated are state of the art. The data on the accumulation of circulating TSP1 in early colitis patients supports these data, while the lack of such effect in severe patients also points to a complex picture, in particular when one considers colitis treatment via the proposed pathway. Notwithstanding, the manuscript combines new mechanistic insight with a perspective for improved patient care.

A few very minor points:

Figure 6 The data in Figure 6C seem to refer to 6B rather than to 6A as indicated in the legends of Figure 6. Please correct, or - if they do indeed refer to Figure 6A explain and provide also the quantitative data for 6B.

Page 23, lines 4 and 16. I must assume that the streptomycin concentration should be 100mg/L or 100 microg/mL.

Page 25. Line 6 Indicate the acetylcholine concentration (probably 100 μ M).

(See next page)

POINT-BY-POINT REPLY

Referee #1 (Remarks for Author):

The submitted MS describes the mechanism of intussusceptive angiogenesis (IA) in inflammatory bowel disease. The authors claim that mice lacking the protease MT1-MMP in endothelial cells (MT1 Δ EC) have limited IA in the capillary plexus of the colon mucosa after 1% DSS-induced colitis. This was correlated with improved tissue perfusion, preserved intestinal morphology, and milder disease activity index. Combined in vivo intravital microscopy and lentiviral rescue experiments with in vitro cell culture demonstrated that MT1-MMP activity in endothelial cells is required for vasodilation and IA, as well as for nitric oxide production via binding of the MT1-MMP substrate thrombospondin-1 (TSP1) to CD47/ α v β 3 integrin. Over all this is a well presented paper that this reviewer had little to add. Experiments are well controlled, not over interpreted and appropriate statistics applied. The mechanism is established. One thing that I missed in Fig 7 was what does 'DAF-FM' stand for in the image analysis?

We appreciate the positive comments and appraisal of our study. We have added the meaning of DAF-FM (4-amino-5-methylamino-2',7'-difluorofluorescein diacetate) in the legend to Figure 4 (the earliest mentioned in the manuscript) as requested by the reviewer.

Referee #2 (Comments on Novelty/Model System for Author):

Major gaps should be filled before the paper is accepted. E.g. TSP1 cleavage is not demonstrated, while the entire manuscript rests on this assumption. The role of intussusceptive angiogenesis remains unclear 0 could be direct effect of TSP1 on macrophages.

Referee #2 (Remarks for Author):

The study by Esteban et al investigates the role of MT1-MMP in ulcerative colitis/ Crohn's disease. The authors link increase MT1 production by intestinal endothelial plexus with increased NO production due to TSP1 cleavage, which hampers TSP1 inhibitor action on NO production and therefore intussusceptive angiogenesis (IA). The study is potentially quite interesting and the concept is novel. However, there are gaps that should be filled prior to publication. In general, manuscript is in need of extensive editing by native English speaker/writer, to eliminate inappropriate word usage. Specific comments are listed below.

We appreciate the reviewer's comments on the interest and novelty of our study. Although a native English speaker already fully edited the original version of our manuscript (as acknowledged), we have made the effort to pay special attention to avoid inappropriate word usage.

1. The description for figure 2 states that the perfusion is inhibited in MT1 delta iEC mice. However, partial perfusion is clearly preserved and this statement should be tempered.

As requested by the reviewer, we have revised the wording relative to Figure 2 in both the text and the legend. We have rephrased the statement as follows: 'Perfusion decreased during colitis progression in the mucosa vascular plexus of both MT1^{f/f} and MT1 ^{Δ EC} mice, but it was significantly better preserved in the latter at 7 days post-1% DSS treatment (Figure 2A-B)'. We think this sentence better describes the results presented in Figure 2 and indicates that indeed MT1 ^{Δ EC} mice maintains partial perfusion as pointed out by the reviewer.

2. Usually vascular abnormalities include vascular leakage (also regulated by TSP) Perfusion using FITC-dextran would answer this question.

Following the reviewer's suggestion, we have analyzed vascular permeability by injecting i.v. directly labeled-dextran. We observed that treatment with 1% DSS induced vascular leakage at day 3 and 7 in the mucosa vascular plexus of MT1^{f/f} mice and at lower extent in MT1^{iΔEC} mice. We have included these new results in the **revised Appendix Figure S2A** and the corresponding text in Results section (page 8, lines 18-20).

3. Fig. 2D shows extensive collagen deposition in F/F mice. However, MT-1 digests multiple collagen types. An explanation should be provided.

As the reviewer points out, it may seem counterintuitive that there is extensive collagen deposition in MT1^{f/f} mice if collagen deposition would just rely on MT1-MMP collagenolytic activity by endothelial cells. It is however important to highlight that in the inflamed intestine the deposition of collagen is mainly related (as in other damaged tissues) to TGFβ pathway activation that in turn leads to MMP inhibition and myofibroblast accumulation (O'Sullivan et al., 2015). Given that colon from DSS-treated MT1^{f/f} mice is more affected compared to MT1^{iΔEC} mice (Figure 2C), it is then expectable an increased collagen accumulation as included in the revised Results section (page 9, line 2). Since MT1-MMP efficiently processes fibrillar collagens type I and III rather than type V which gives an intense SHG signal, our data would also point to collagen type V as one the most prominent collagens accumulated in colitis-inflamed intestine as previously suggested (Petrey and de la Motte, 2017; Ajeti et al., 2011).

4. The differences in IA are clear after 3 days, however there is no clear differences in intestinal morphology and macrophage recruitment. Is it possible that IA is not the driving force behind DCC-induced colitis but merely a correlate? Could IA in MT1 delta iEC mice be restored using exogenous angiogenic factor, e.g. VEGF? A propos, it can be released from the ECM by MT1-MMP. Also, TSP1 can have a direct effect on macrophage recruitment and inflammation, via CD36. These possibilities should be discussed in some depth.

Indeed, we consider an advantage of 1% DSS model that 3 days after treatment there are not major alterations in intestinal histology and leukocyte recruitment what allowed us observe a significant reduction of IA in MT1^{iΔEC} mice at this time point. This finding argues in favor of IA as one the earliest pathological events in this model of mild colitis instead of being secondary to tissue inflammation and damage as discussed (page 16, lines 22-25). In relation to the possible rescue of IA by VEGF in MT1^{iΔEC} mice, we detected similar VEGF serum levels in 1% DSS-treated MT1^{iΔEC} mice compared to MT1^{f/f} controls (VEGF_{MT1^{iΔEC}} = 84.27 pg/ml and VEGF_{MT1^{f/f}} = 85.22 pg/ml) suggesting that differences in VEGF availability will not be in principle influencing the IA phenotype in the MT1-MMP-deficient mice. This together with our new data (see below) showing that VEGF-induced vasodilation in mucosa capillaries is reduced in MT1-MMP absence from ECs make us posit that restoration of IA in MT1^{iΔEC} mice with exogenous VEGF seems unlikely. We fully agree with the reviewer on the functional pleiotropism of TSP1 in colitis, in particular in promoting inflammatory cell infiltration. However, since CD36 binding site in TSP1 would be disrupted after MT1-MMP cleavage and we observe only a minor inflammatory component our data suggest a major contribution of MT1-MMP-processed TSP1 to IA via CD47/αvβ3 integrin rather than to macrophage recruitment and inflammation via CD36 during 1% DSS-induced mild colitis. We mention this point in the revised Discussion (page 20, lines 11-13).

5. AcCh is a bit drastic for induction of vasodilation. VEGF would be a more physiological agent. Vasodilation in cremaster muscle is not quite reflective of IA in the colon.

We appreciate this reviewer's comment and we agree VEGF is a more physiological stimulus of vasodilation than AcCh and that cremaster muscle does not recapitulate the capillary architecture of the colon mucosa. Therefore, we have implemented local application of VEGF with a rectal cannula to the distal colon and assessed vasodilation of the mucosal plexus to get closer to the pathophysiology of IA during colitis. We first confirmed that VEGF induced vasodilation of the capillaries in the mucosal plexus (from a diameter of 5 μ m in control to averaged 6.25 μ m in VEGF-treated colons). Then we quantified the vasodilation response in MT1^{f/f} and MT1^{i Δ EC} mice and we observed a significant reduction of the capillary diameter in the mucosal plexus of VEGF-treated MT1^{i Δ EC} mice compared to MT1^{f/f} controls (**new Figure EV4**; page 10, lines 18-22). These new data expand the requirement of MT1-MMP for vasodilation induced by the physiological stimulus VEGF (which also activates nitric oxide production; Cooke and Losordo 2002) in the intestinal mucosa vasculature prior to IA during colitis as discussed in the revised version (page 17, lines 11-14).

6. *The effects of MT1 antibodies and TSP1 receptors on NO production are quite clear. Would MT1 antibody and TSP1 derived ligands alter IA as well?*

As requested by the reviewer we have checked the effect on IA of the inhibitory anti-MT1-MMP LEM-2/15 mAb (Sagi et al., 2017). As shown in **revised Figure 8C-E**, intraperitoneal injection of the LEM-2/15 mAb in MT1^{f/f} mice decreases the number of IA events and also preserves collagen fiber organization compared to the isotype control after 3 days of 1% DSS treatment (page 15, lines 10-15), recapitulating the phenotype observed in MT1^{i Δ EC} mice.

About the effect of TSP1-derived ligands on IA, we already demonstrated that blocking TSP1 interaction with the α v β 3 integrin by the nonamer peptide GDGRGDACK decreased IA *in vivo* in MT1^{f/f} mice, and previous reports had shown that blocking CD47 binding to TSP1 improved colitis in IL10-null mice but not in DSS-induced model (Danese et al., 2007) as discussed. As requested, we have evaluated now the effect of the whole TSP1 fragment predicted to be generated by MT1-MMP cleavage and thus to stimulate α v β 3 integrin-CD47/nitric oxide/IA pathway. Our collaborator Giulia Taraboletti (Istituto di Ricerche Farmacologiche Mario Negri IRCCS, Bergamo, Italy) provided us the recombinant E123CaG-1 human TSP1 (549-1170) spanning the expected fragment derived from MT1-MMP cleavage (441/467-1170) and containing the binding sites for α v β 3 integrin-CD47 (Margosio et al., 2008). We tested the effect of this fragment and observed that it increased nitric oxide production *in vitro* in siMT1-HUVEC compared to full TSP1 and when administered intraperitoneally it also boosted the number of IA events *in vivo* in MT1^{i Δ EC} mice compared to full length TSP1 (**revised Figure EV5D-F**; page 13, lines 10-16 and page 15, lines 6-10).

7. *Fig. 6A-C - the lack of TSP1 in the vasculature is an overstatement: TSP1 is clearly detectable and the images do not convey major differences. Co-localization with MT1 would be helpful in f/f mice.*

We regret our confusing wording of the results presented in Fig 6A-C. For clarification we have rephrased the results as follows: 'Immunostained sections revealed that 1% DSS treatment upregulated TSP1 expression in the mouse colon after 3 days (**Figure 6A-B**). Moreover, image analysis demonstrated that TSP1 was significantly accumulated forming a thick cuff around MT1-MMP-negative vessels of the colon mucosa nearby the muscularis mucosa from MT1^{i Δ EC} mice compared with less TSP1 abundance around the corresponding MT1-MMP-expressing vessels in MT1^{f/f} mice (**Figure 6A-C**). These data indicate that endothelial MT1-MMP contributes to perivascular TSP1 processing in the intestine.' There are no differences in TSP1 localization but rather on its perivascular

abundance as shown in Figure 6C. Moreover, as requested by the reviewer, we have double-stained MT1-MMP and TSP1 to visualize their co-expression in vessels. This approach was not performed in the original version due to incompatibility in the species of primary antibodies (both rabbit). We have solved this limitation by using the mouse anti-MT1-MMP antibody (LEM-2/15). As shown in **revised Figure 6B**, MT1-MMP is expressed in endothelial cells of some large vessels of the deep mucosa, which are surrounded by TSP1. Noteworthy, MT1-MMP-negative vessels in MT1-MMP^{iAEC} mice display a thicker TSP1-positive cuff in relation to its impaired processing in the absence of MT1-MMP in endothelial cells as stated in the revised version. It is interesting to mention that in the colon mucosa of MT1^{f/f} mice we could find a few vessels negative for MT1-MMP; in these particular vessels, the surrounding TSP1 cuff was thicker than in MT1-MMP-expressing vessels (**Figure 1 for reviewers**). This observation further supports the contribution of endothelial MT1-MMP to TSP1 perivascular processing and to its accumulation when MT1-MMP is absent from endothelial cells.

Figure 1. TSP1 accumulates around MT1-MMP-negative vessels in the intestinal mucosa. Representative maximum intensity images of CD31 (gray), TSP1 (red), and MT1-MMP (green) in colon from MT1^{f/f} control mice 3 days after 1% DSS treatment. Note that the perivascular TSP1 cuff is thicker in those vessels in which MT1-MMP expression is low or absent (two lower rows) compared to MT1-MMP-expressing vessels (two upper rows). Scale bar, 10 μ m.

8. To demonstrate TSP1 cleavage, it would be helpful to incubate intact TSP1 with MT1 positive and negative EC and run a western blot with antibodies to C and N-termini. Otherwise this remains a mere speculation.

Chris Overall's group previously demonstrated TSP1 cleavage by MT1-MMP by *in vitro* digestion assays and two main fragments (~ 100 and 50 kDa) were observed (Butler et al., 2008) and we identified TSP1 as a substrate for MT1-MMP in TNF α -stimulated endothelial cells by SILAC (Koziol et al., 2012). Nevertheless the proteolytic site for MT1-MMP in TSP1 remained undefined and that is why in the present study we performed *in silico* prediction and modelling of putative cleavage sites. As requested by the reviewer, we have assessed the TSP1 cleavage sites by endothelial MT1-MMP analyzing TSP1 fragments in lysates from non-treated siCtrl (MT1-MMP-positive) and siMT1-silenced (MT1-MMP-negative) ECs (since we already observed different nitric oxide production in these conditions, **Figure 4A**). As shown in new **Figure EV5A**, Western blot with the antibody recognizing an epitope upstream of the predicted TSP1 cleavage sites for MT1-MMP (H⁴⁴¹W and P⁴⁶⁷Q), revealed an N-terminal fragment of about 50 kDa consistent with the predicted cleavage sites and whose abundance was reduced in siMT1-silenced ECs. Complementarily, *in vitro* digestion assays with full-length TSP1 and human recombinant MT1-MMP catalytic domain confirmed the dose-response generation of the 50 kDa N-terminal TSP1 fragment by MT1-MMP (new **Figure EV5B**). Unfortunately, the antibody against the TSP1 fragment downstream of MT1-MMP cleavage (A6.1, epitope in the first calcium-binding motif, 692–717) worked with low sensitivity by Western blot (Annis et al., 2006), and it was able to barely detect full-length TSP1 but not any C-terminal fragment in lysates and *in vitro* digestion assays (**Figure 2 for reviewers**). These findings together with our new data showing that the C-terminal TSP1 fragment produces nitric oxide *in vitro* and IA *in vivo* in MT1-MMP-deficient endothelial cells demonstrate the pathological relevance of this processing.

Figure 2. Analysis of TSP1 fragments in HUVEC lysates with the anti-TSP1 antibody A6.1. Western blot of TSP1 expression (developed with the anti-TSP1 antibody A6.1 against an epitope downstream of the TSP1 cleavage site for MT1-MMP) in lysates from siCtrl and siMT1-silenced HUVEC. MT1-MMP and tubulin are included as silencing and loading controls (A); recombinant human (rh)MT1-MMP catalytic domain is also shown (B). The arrowhead marks full-length TSP1. No C-terminal TSP1 fragments can be detected with this antibody.

9. Fig. 8, MT1 MMP staining looks unconvincing. Improved quality images are required.

We regret the suboptimal quality of the original images presented in original Figure 8A. In this revised version, we have provided better quality images that show clearer MT1-MMP staining, particularly in the vessels of the inflamed colonic tissue from IBD patients (see merged image with CD31).

10. Supplemental Fig. 8; CD36 binding peptide sequence is GVITRIR, not GNITRIR.

We regret this mistake and we have corrected the sequence in the revised version of the manuscript.

Referee #3 (Comments on Novelty/Model System for Author):

The manuscript deals with a clinical important issue. The involvement of the MT1-MMP-TSP1-NO pathway is novel. The techniques are state of the art.

The authors use a combination of existing animal models, deletion mutant, lentiviral constitution of MT1-MMP and novel point mutants in MT1-MMP to underpin their findings.

The molecular mechanisms of anti-TNF α antibodies in inflammatory bowel disease are still a matter of debate. The present manuscript provides detailed information how TNF α -mediated induction of MT1-MMP via thrombospondin and NO generation affects vasodilation of capillaries accompanied by intussusceptive angiogenesis.

The manuscript has merit for both better understanding of altered vascularization in colitis and is a welcome contribution increasing our insight in the regulation of intussusceptive angiogenesis in general. Although this pathway may not the only pathway that is involved in this disease, the authors make plausible that the proposed MT1-MMP-TSP1-integrin $\alpha\beta 3$ -NO pathway is a prerequisite for intussusceptive angiogenesis, and contributes to colitis. The techniques used and new mutants generated are state of the art. The data on the accumulation of circulating TSP1 in early colitis patients supports these data, while the lack of such effect in severe patients also points to a complex picture, in particular when one considers colitis treatment via the proposed pathway. Notwithstanding, the manuscript combines new mechanistic insight with a perspective for improved patient care.

I had only a few very minor remarks and would like to recommend the manuscript or publication in EMBO Molecular Medicine.

Referee #3 (Remarks for Author):

The molecular mechanisms of anti-TNF α antibodies in inflammatory bowel disease are still a matter of debate. The present manuscript provides detailed information how TNF α -mediated induction of MT1-MMP via thrombospondin and NO generation affects vasodilation of capillaries accompanied by intussusceptive angiogenesis. The manuscript has merit for both better understanding of altered vascularization in colitis and is a welcome contribution increasing our insight in the regulation of intussusceptive angiogenesis in general. Although this pathway may not the only pathway that is involved in this disease, the authors make plausible that the proposed MT1-MMP-TSP1-integrin $\alpha\beta 3$ -NO pathway is a prerequisite for intussusceptive angiogenesis, which contributes to colitis. The techniques used and new mutants generated are state of the art. The data on the accumulation of circulating TSP1 in early colitis patients supports these data, while the lack of such effect in severe patients also points to a complex picture, in particular when one considers colitis treatment via the proposed pathway. Notwithstanding, the manuscript combines new mechanistic insight with a perspective for improved patient care.

A few very minor points:

Figure 6 The data in Figure 6C seem to refer to 6B rather than to 6A as indicated in the legends of Figure 6. Please correct, or - if they do indeed refer to Figure 6A explain and provide also the quantitative data for 6B.

We are sorry for the confusing wording. Quantification in Figure 6C refers to analysis of vessels from 1% DSS-treated mice presented in Figure 6B. We have corrected the legend accordingly.

Page 23, lines 4 and 16. I must assume that the streptomycine concentration should be 100mg/L or 100 microg/mL.

We regret the mistake and we have corrected streptomycin concentration to 100 μ g/ml.

Page 25. Line 6 Indicate the acetylcholine concentration (probably 100 μ M).

We appreciate the reviewer's comment and we have added the concentration of acetylcholine (100 μ M) as requested.

2nd Editorial Decision

25 October 2019

Thank you for the submission of your revised manuscript to EMBO Molecular Medicine. We have received the referees' reports, and as you will see they are now supportive of publication of your study. I am therefore pleased to inform you that we will be able to accept your manuscript pending minor editorial amendments and a response to the minor comment from referee #3.

I look forward to reading a new revised version of your manuscript as soon as possible.

***** Reviewer's comments *****

Referee #1 (Comments on Novelty/Model System for Author):

The work is very well done and all revisions have been made adequately.

Referee #1 (Remarks for Author):

Congratulations to the authors for producing an interesting and well done set of experiments.

Referee #2 (Remarks for Author):

All the comments were thoroughly addressed, excellent study and superb work.

Referee #3 (Comments on Novelty/Model System for Author):

The manuscript deals with a clinical important issue.
The involvement of the MT1-MMP-TSP1-NO pathway is novel.
The techniques are state of the art.

Referee #3 (Remarks for Author):

The additional data further strengthen the manuscript.
My questions have been adequately answered.

Very minor remark:

Introduction page 1 : "Nevertheless, genes whose expression is enriched during IA have been identified in skeletal muscle, in which the vasodilator prazosin induces IA and the excision of the agonist muscle, sprouting angiogenesis (Egginton, 2011)."

Maybe I missed it, but I could not find back this information on IA in the review referred to, but did find such prazosin data in [Zhou, Egginton et al, 1998 Cell Tissue Res. 1998 Aug;293(2):293-303.]
The authors may wish to check the reference. (Egginton wrote several reviews in 2011.)

2nd Revision - authors' response

5 November 2019

We have amended the minor comment from reviewer 3 by adding the suggested reference in the corresponding text section and revised the Main manuscript file as recommended. We have modified the Figures and Appendix as requested. We have chosen to include the exact p-values for all panels/figures in the supplemental Table S3 in the Appendix. However, we have not replaced the bar graphs by scatter plots, in spite of agreeing they are more informative, for the sake of time and because individual data points are now available in the provided Source data file.

Corresponding Author Name: Alicia G. Arroyo
 Journal Submitted to: EMBO Molecular Medicine
 Manuscript Number: EMM-2019-10862-V2